# Balanced Learning for Domain Adaptive Semantic Segmentation

**Wangkai Li** [1]  **Rui Sun** [1]  **Bohao Liao** [1]  **Zhaoyang Li** [1]  **Tianzhu Zhang** [1] [2]

## Abstract

Unsupervised domain adaptation (UDA) for semantic segmentation aims to transfer knowledge from a labeled source domain to an unlabeled target domain. Despite the effectiveness of self-training techniques in UDA, they struggle to learn each class in a balanced manner due to inherent class imbalance and distribution shift in both data and label space between domains. To address this issue, we propose Balanced Learning for Domain Adaptation (BLDA), a novel approach to directly assess and alleviate class bias without requiring prior knowledge about the distribution shift. First, we identify over-predicted and under-predicted classes by analyzing the distribution of predicted logits. Subsequently, we introduce a post-hoc approach to align the logits distributions across different classes using shared anchor distributions. To further consider the network's need to generate unbiased pseudo-labels during self-training, we estimate logits distributions online and incorporate logits correction terms into the loss function. Moreover, we leverage the resulting cumulative density as domain-shared structural knowledge to connect the source and target domains. Extensive experiments on two standard UDA semantic segmentation benchmarks demonstrate that BLDA consistently improves performance, especially for under-predicted classes, when integrated into various existing methods. Code is available at https://github.com/Woof6/BLDA.

## 1. Introduction

Semantic segmentation, which assigns semantic labels to each pixel in an image, has made remarkable progress in recent years (Long et al., 2015a; Chen et al., 2017b; Cheng

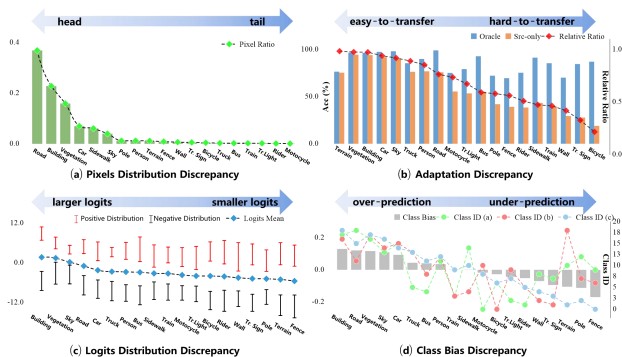

Figure 1: Demonstration of factors that cause class bias. (a) The inherent class imbalance problem in segmentation datasets. (b) The differences in transfer difficulty across classes in cross-domain settings. "Oracle' represents the performance under full supervision, while "Src-only' represents training with the source domain and testing it on the target domain. (c) The differences in logits distributions predicted for each class, including "positive distribution" and "negative distribution". (d) Bias assessment for different classes via Eq.4. The corresponding class IDs of (a), (b), and (c) are mapped in descending order onto this figure.

et al., 2021; 2022). However, the performance of these methods often degrades significantly when applied to new target domains due to differences between the source and target domains. Unsupervised domain adaptation (UDA) techniques have been extensively studied to address this issue by transferring knowledge from a labeled source domain to an unlabeled target domain, aiming to bridge the domain gap and improve the model's performance on the target dataset without requiring additional annotations.

In previous work, self-training techniques (Tranheden et al., 2021; Hoyer et al., 2022a) have been naturally introduced into UDA tasks to fully utilize the large amount of unlabeled target domain data, becoming a mainstream paradigm. This paradigm constructs a teacher network using a temporal aggregation mechanism, treats its predictions on the target domain as pseudo-labels, and gradually guides the student network's learning. Despite achieving remarkable results, these methods struggle to learn each class in a balanced manner. Generally, the inherent class imbalance in segmentation datasets (Cordts et al., 2016) (Fig.1(a)) leads networks

---

[1]MoE Key Laboratory of Brain-inspired Intelligent Perception and Cognition, University of Science and Technology of China [2]Deep Space Exploration Laboratory. Correspondence to: Tianzhu Zhang <tzzhang@ustc.edu.cn>.

*Proceedings of the 42nd International Conference on Machine Learning*, Vancouver, Canada. PMLR 267, 2025. Copyright 2025 by the author(s).

to produce biased predictions towards head classes, often studied as the long-tail problem (Van Horn & Perona, 2017; Buda et al., 2018; Liu et al., 2019). However, in UDA, data and label distribution shifts between the training and test data complicate the class bias. The network's bias towards classes does not entirely depend on the differences in class sample distribution. As shown in Fig.1(b), when a network trained on the source domain is tested on the target domain, the performance degradation varies greatly across classes, distinguishing easy-to-transfer and hard-to-transfer classes. These factors jointly determine the network's different biases towards each class in target domain, resulting in over-prediction and under-prediction. Furthermore, confirmation bias (Guo et al., 2017) causes self-training techniques to exacerbate this phenomenon. Fig.2(a) shows the severe deterioration of classes like *rider* and *bicycle* after self-training, widening the performance gap across classes. Therefore, achieving balanced learning for each class in UDA is a challenging and worthwhile exploration.

Existing strategies to reduce model bias towards different classes can be broadly categorized into re-weighting (Cui et al., 2019; Lin et al., 2017; Cao et al., 2019; Truong et al., 2023; Buda et al., 2018) and re-sampling (Hoyer et al., 2022a; Araslanov & Roth, 2021; He et al., 2008; 2021; Guan et al., 2022). To compare these methods, we take self-training as the baseline method in Fig.2 and implement re-weighting (Cui et al., 2019) and re-sampling (Hoyer et al., 2022a) techniques, respectively. We observe the class bias through class-wise accuracy (Fig.2(a)) and the frequency of pseudo-labels generated on the target domain during training (Fig.2(b)). Loss re-weighting aims to assign different weights to classes, making the model pay more attention to tail classes. Although intuitive, the update frequency of each class to the network still varies greatly, with some classes remaining challenging to learn effectively, resulting in unstable performance in self-training. In contrast, sample re-sampling proves more effective by directly adjusting the class sample distribution during training, significantly enhancing the performance of tail classes. Despite their empirical solid performance, these methods are heuristic and rely on the assumption that the test and training data share the same distribution in both data and label space. However, in the UDA setting, these assumptions are invalid because (1) the class distributions of the source and target domains differ, and the target domain's prior class distribution is unavailable; (2) the data distributions also differ, leading to varying transfer difficulties across classes in cross-domain settings. This raises the question: *How to assess and alleviate class bias directly without requiring prior knowledge about the distribution shift between the two domains?*

In this work, we propose to assess the degree of class bias by analyzing the distribution of logits predicted by the network (Sec.3.3.2). Fig.1(c) shows that the network exhibits

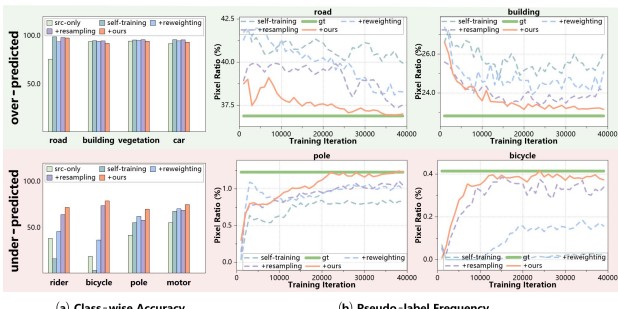

(a) Class-wise Accuracy      (b) Pseudo-label Frequency

Figure 2: In UDA, class bias can be expressed as over-predicted classes and under-predicted classes. (a) Class-wise accuracy under different training settings. (b) Frequency of pseudo-labels generated by the network for different classes during training.

differences in the predicted logits distributions for different classes, directly leading to class bias. Fig.1(d) illustrates that the ranking of class bias highly coincides with the ranking of logit distribution differences, i.e., over-predicted classes have larger logit values, while under-predicted classes have smaller logit values. This assessment approach prompts us to propose BLDA, a method to achieve balanced learning for domain adaptive semantic segmentation by balancing the logits distribution. First, we consider a post-hoc approach to adjust the logits (Sec.3.3.3). We set shared anchor distributions for the positive and negative logits distributions and align the class-wise logits distributions with the anchor distributions based on the cumulative density function mapping. Furthermore, to generate unbiased pseudo-labels for classes during self-training, we propose an online logit adjustment method (Sec.3.3.4). This strategy couples Gaussian mixture models to estimate the logits distributions online during training and incorporates logit correction terms into the loss function to replace the post-hoc method. Moreover, we find that the resulting cumulative density can measure the discrimination difficulty of different sample points in each class, which is a domain-shared structural knowledge that can be used as an auxiliary loss to connect the two domains, further enhancing domain adaptation performance (Sec.3.3.5). As shown in Fig.2, our method can be integrated into existing self-training-based UDA paradigms and effectively balance the prediction bias across classes.

Our contributions can be summarized as follows: (1) **Problem Identification** (*Why*): We provide a comprehensive analysis of class bias in UDA settings, revealing that it is jointly affected by distribution shifts in both label and data spaces, making it difficult to mitigate using priors like regular class-imbalanced problems. (2) **Methodology Design** (*What*): We propose a new perspective to assess and effectively mitigate class bias by designing a class balanced learning strategy that estimates correction terms from logits distribution, addressing the limitations of existing class-

imbalanced solutions. (3) **Implementation** (*How*): We implement our strategy through a plug-and-play module with broad application potential in the UDA field. It first implements distribution estimation, then applies an online logits adjustment to perceive the model's learning state, achieving class-balanced learning. (4) **Extensive Experiments**: We validate our approach through extensive experiments across various UDA benchmarks, tasks, and architectures, demonstrating that class bias is a widespread challenge in UDA. Our method achieves consistent and significant performance gains, showcasing its effectiveness and versatility.

## 2. Related Work

Here, we provide a brief overview of the related work. For a more detailed discussion, please refer to Appendix P.

### 2.1. Domain Adaptive Semantic Segmentation

Unsupervised domain adaptation (UDA) aims to transfer semantic knowledge from labeled source domains to unlabeled target domains, and is crucial for semantic segmentation to avoid laborious pixel-wise annotation in new target scenarios. Recent UDA approaches for semantic segmentation can be categorized into two main paradigms: adversarial training-based methods (Toldo et al., 2020; Tsai et al., 2018; Chen et al., 2018; Ganin & Lempitsky, 2015; Hong et al., 2018; Long et al., 2015b) and self-training-based methods (Tranheden et al., 2021; Hoyer et al., 2022a; Araslanov & Roth, 2021; Zhang et al., 2021). Adversarial training-based methods learn domain-invariant representations through a min-max optimization game, where a feature extractor is trained to confuse a domain discriminator, aligning feature distributions across domains. Self-training-based methods, which have come to dominate the field due to the domain-robustness of Transformers (Bhojanapalli et al., 2021), generate pseudo labels for target images based on a teacher-student optimization framework. However, due to the inherent class imbalance and distribution shift in both data and label space between domains, networks often produce complicated class bias, which is further exacerbated by confirmation bias in the self-training paradigm. Our method focuses on balanced learning to address these unique challenges in UDA training.

### 2.2. Class-Imbalanced Learning

Class imbalance is a common problem in semantic segmentation, where the number of samples per class varies significantly. Existing methods address this issue through re-weighting or re-sampling techniques. Re-weighting methods assign different weights to classes during training, giving higher importance to under-represented classes (Cui et al., 2019; Lin et al., 2017; Cao et al., 2019). Re-sampling techniques modify the class distribution in the training data

by over-sampling minority classes or under-sampling majority classes (He et al., 2008; 2021). In UDA for semantic segmentation, several approaches have introduced these strategies to alleviate class bias (Hoyer et al., 2022a; Araslanov & Roth, 2021; Li et al., 2022). However, these methods are still empirical and focus on the single-domain setting, which follows the assumptions that the test data and training data share the same distribution in both data space and label space, without considering the additional challenges posed by domain shift in UDA. In this work, we aim to access class bias directly and achieve balanced learning for each class with no prior knowledge about the distribution shift between domains.

## 3. Method

### 3.1. Problem Definition

In unsupervised domain adaptation for semantic segmentation, the network is simultaneously trained on labeled source domain data and unlabeled target domain data. To be specific, the source domain can be denoted as $D_S = \{(x_i^S, y_i^S)\}_{i=1}^{N_S}$, where $x_i^S \in X_S$ represents an image with $y_i^S \in Y_S$ as the corresponding pixel-wise one-hot label covering $C$ classes. The target domain can be denoted as $D_T = \{(x_i^T)\}_{i=1}^{N_T}$, which shares the same label space but has no access to the target label $Y_T$.

### 3.2. Revisiting Self-training in UDA

Self-training-based pipelines for UDA segmentation consist of a supervised branch for the source domain and an unsupervised branch for the target domain. For the supervised branch, loss $\mathcal{L}^s$ can only be calculated on the source domain to train a neural network $f_\theta$:

$$\mathcal{L}^s = \frac{1}{N_S} \sum_{i=1}^{N_S} \frac{1}{HW} \sum_{j=1}^{H \times W} \ell_{ce}(f_\theta(x_{ij}^S), y_{ij}^S), \quad (1)$$

where $\ell_{ce}$ denotes the cross-entropy loss. Unsupervised branch introduces teacher-student framework to generate pseudo-labels $\hat{y}_{ij}^T = \text{argmax}(g_\phi(x_{ij}^T))$ with the teacher model $g_\phi$ for target domain:

$$\mathcal{L}^u = \frac{1}{N_T} \sum_{i=1}^{N_T} \frac{1}{HW} \sum_{j=1}^{H \times W} q(p_{ij})\ell_{ce}(f_\theta(x_{ij}^T), \hat{y}_{ij}^T), \quad (2)$$

where we define $q(p_{ij})$ as a quality estimate conditioned on confidence $p_{ij} = max(g_\phi(x_{ij}^T))$ for pseudo labels, which gradually strengthens with increasing accuracy of models and can be implemented with threshold filtering or a weighting function. After each training step, the teacher model $g_\phi$ is updated with the exponentially moving average of the weights of $f_\theta$. Then, the overall objective function is a combination of supervised loss and unsupervised loss as $\mathcal{L} = \mathcal{L}^s + \mathcal{L}^u$.

### 3.3. Balanced Learning for Domain Adaptive Semantic Segmentation

#### 3.3.1. OVERVIEW

In this section, we first propose to assess the network's prediction bias towards each class by statistically analyzing the distribution of logits (Sec.3.3.2). Based on the above analysis, we define a post-hoc method to balance the network's predictions (Sec.3.3.3). Furthermore, we introduce online logits adjustment tailored for the UDA training process (Sec.3.3.4). Finally, we introduce cumulative density estimation as domain-shared knowledge to bridge the two domains (Sec.3.3.5).

#### 3.3.2. ASSESSING PREDICTION BIAS FROM LOGITS DISTRIBUTION

Given a label space $\mathcal{Y} = [C] = \{1, 2, ..., C\}$, the segmentation network can be seen as a scorer $f_\theta : x_{ij} \to \mathcal{R}^C$ that assigns class-wise scores, also known as logits, to a pixel $x_{ij}$ from image $x_i$. To investigate the distribution of logits obtained by the network for different classes, we can analyze it from the perspective of the confusion matrix. The confusion matrix is a $C \times C$ matrix $M$, where each element $M_{cl}$ represents the number of pixels with ground truth label $c$ predicted as class $l$. We replace each element $M_{cl}$ in the confusion matrix $M$ with the corresponding set of logits, i.e., the logits predicted for class $l$ for all pixels with ground truth label $c$, to obtain the logits set matrix $\mathcal{M}$. We then use $\mathcal{M}$ to assess prediction bias.

**Definition 1.** Element in logits set matrix, $\mathcal{M}_{cl}$.

$$\mathcal{M}_{cl} = \{f_\theta(x_{ij})[l] \mid y_{ij} = c\}, \qquad (3)$$

where $f_\theta(x_{ij})[l]$ represents the logit value predicted by the network for class $l$ of pixel $x_{ij}$, and $y_{ij}$ is the ground truth. In the resulting $C \times C$ matrix $\mathcal{M}$, diagonal elements $\mathcal{M}_{ll}$ represent the "positive logits distribution" for class $l$, while off-diagonal elements $\mathcal{M}_{cl}$ ($c \neq l$) represent the "negative logits distribution" for class $l$ with respect to class $c$.

**Definition 2.** Bias of the network towards class $l$, $\text{Bias}(l)$.

$$\text{Bias}(l) = \frac{1}{C} \sum_{c \in [C]} \mathbb{P}(\arg\max_{c' \in [C]} f_\theta(x)[c'] = l|y = c) - \frac{1}{C}, \quad (4)$$

where $\mathbb{P}(\arg\max_{c' \in [C]} f_\theta(x)[c'] = l|y = c)$ represents the probability that the network predicts a sample from class $c$ as class $l$. This definition measures the average difference between the probability of predicting class $l$ and the uniform probability $1/C$ across all classes. A positive bias indicates over-prediction, while a negative bias indicates under-prediction for class $l$.

Let $P_{cl}$ denote the distribution of $\mathcal{M}_{cl}$. Assuming each distribution $P_{cl}$ is independent, we can estimate the prediction

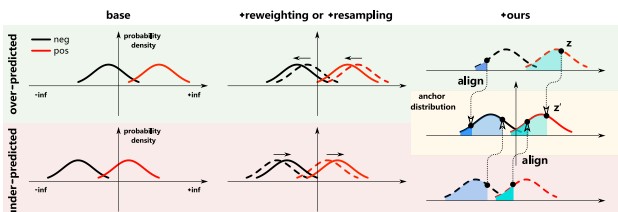

Figure 3: Illustration of proposed post-hoc class balancing. (a) The logits distributions of over-predicted and under-predicted classes. (b) Reweighting/resampling strategies alleviate class imbalance by adjusting the training emphasis on different classes. (c) Our post-hoc logits adjustment method aligns the logits distributions of all classes with anchor distributions to achieve balanced prediction.

probability in Eq.4 by comparing the logit values:

$$\mathbb{P}(l|c) \approx \int_{-\infty}^{\infty} P_{cl}(z) \prod_{y' \neq l} \left( \int_{-\infty}^{z} P_{cy'}(t)dt \right) dz. \qquad (5)$$

Combining Eq.4 and Eq.5, for an unbiased network, i.e., $\text{Bias}(l) = 0$ for all $l \in [C]$, a sufficient condition is that they have the same positive and negative distributions. This means the network's prediction performance is consistent across all classes. Fig.1(d) shows a direct correlation between logit distribution differences and class bias, indicating that variations in logit distributions lead to class bias in the network's predictions.

#### 3.3.3. POST-HOC CLASS BALANCING

Generally, the network tends to produce larger logits for over-predicted classes and smaller logits for under-predicted classes, as shown in Fig.3(a). Reweighting/resampling strategies can alleviate this gap by making the network pay more attention to tail classes during training, as illustrated in Fig.3(b). However, as shown in Fig.1, class bias does not fully correlate with the inherent class imbalance problem, especially in the UDA setting, where different distribution shifts exist in both data and label space between domains. Furthermore, these methods are empirical and lack generalization capability across various scenarios.

Based on the above analysis, to balance the network's prediction capabilities across classes, we adjust the network's predictions in a post-hoc manner. Specifically, we define an anchor distribution $P_p$ for the positive logits distribution and an anchor distribution $P_n$ for the negative logits distribution. We then align all the logits distributions with corresponding anchor distributions, as shown in Fig.3(c). To preserve the relative ordering of logits, i.e., structural information within each distribution, we align them in a point-wise way via the cumulative distribution function (CDF). Let $F_{cl}(z)$, $F_p(z)$ and $F_n(z)$ be the CDFs of $P_{cl}$, $P_p$ and $P_n$, respectively. We

align the logit value $z$ from $P_{cl}$ to $P_p$ or $P_n$ as follows:

$$z' = \begin{cases} F_p^{-1}(F_{cl}(z)), & \text{if } c = l \\ F_n^{-1}(F_{cl}(z)), & \text{if } c \neq l \end{cases} \quad (6)$$

where $z'$ is the aligned value of $z$ with respect to the anchor distribution. For brevity, we define an offset for logit as $\Delta_{cl}(z) = z' - z$. Considering the probability estimate that $p(y_{ij} = c|x_{ij}) \propto \exp(f_\theta(x_{ij})[c])$, we can obtain revised prediction results for each pixel $x_{ij}$ by:

$$\widetilde{y}_{ij} = \arg\max_{c \in [C]} \frac{\exp(f_\theta(x_{ij})[c] + \tau\Delta_{cc}(f_\theta(x_{ij})[c]))}{\sum_{c' \in [C]} \exp(f_\theta(x_{ij})[c'] + \tau\Delta_{cc'}(f_\theta(x_{ij})[c']))}, \quad (7)$$

where $\tau$ is a scaling factor (the derivation of Eq.7 is detailed in Appendix A). When $\tau = 1$, the model produces balanced predictions. However, to achieve optimal performance on specific evaluation metrics (e.g., mIoU), we need to adjust the value of $\tau$. We discuss the choice of $\tau$ in detail in the experimental section. In this way, the network generates balanced predictions for different classes.

**Discussion about anchor distribution.** In our experiments, we use the global positive and negative logits distributions on the source domain to estimate the anchor distribution for both source and target domains. This choice is based on two key considerations: (1) The network tends to produce larger logits for over-predicted classes and smaller logits for under-predicted classes. By using the global logits distribution, we can effectively measure the average learning degree of the network across all classes. Aligning each class-specific distribution with this global distribution can help neutralize the class bias of the network, ensuring a more balanced learning process. (2) When estimating the logits distributions for each class, there exist varying degrees of statistical errors. According to Bernstein inequalities, estimating the global logits distribution can reduce estimation errors to a certain extent and accelerate convergence rates. In our case, the global distribution, being more robust and stable, serves as a reliable anchor distribution for subsequent alignment. We provide further analysis and discussion in Appendnxi J.

### 3.3.4. ONLINE LOGITS ADJUSTMENT FOR UDA

While the above post-hoc logits adjustment method can effectively balance the network predictions across different classes, it is performed after training the model. In the UDA setting, it is significant to incorporate the logits balancing mechanism directly into the training process. By doing so, the model can learn to make more balanced predictions while adapting to the target domain through pseudo labels.

To achieve this, we propose an online logit adjustment method tailored for UDA training. The key to this method lies in the online estimation of the logits distributions. We employ Gaussian Mixture Models (GMMs) to model these

distributions. Considering that the source and target domains have inherently different logits distributions, we maintain two sets of GMMs separately, with each set containing $C \times C \times K$ Gaussian components, where $C$ denotes the number of classes and $K$ denotes the number of Gaussian components per element in $\mathcal{M}$. Formally, we define:

$$P_{cl}^S = \sum_{k=1}^{K} \pi_{clk}^S \mathcal{N}(\mu_{clk}^S, \sigma_{clk}^S), \quad P_{cl}^T = \sum_{k=1}^{K} \pi_{clk}^T \mathcal{N}(\mu_{clk}^T, \sigma_{clk}^T),$$
$$(8)$$

where $P_{cl}^S$ and $P_{cl}^T$ represent the estimated logits distributions for class $l$ when the ground truth label or pseudo label is $c$ in the source and target domains, respectively. The parameters $\pi_{clk}^S$, $\mu_{clk}^S$, $\sigma_{clk}^S$ and $\pi_{clk}^T$, $\mu_{clk}^T$, $\sigma_{clk}^T$ denote the mixing coefficient, mean, and standard deviation of the $k$-th Gaussian component in the corresponding GMM. We update the GMM parameters during each training iteration using the logits obtained from the current mini-batch via the Expectation-Maximization (EM) algorithm (McLachlan & Krishnan, 2007). Since the network $f_\theta$ gradually evolves during training, we adopt a momentum-based EM update strategy. Specifically, we directly use the GMM parameters $\hat{\phi}_{cl}$ estimated in the latest iteration as the initialization $\phi_{cl}^{(0)}$ for the current iteration. After $\mathcal{T}$ EM loops, the current iteration is completed, and a momentum update is adapted with $\phi_{cl}^{(\mathcal{T})} \leftarrow (1 - \tilde{\tau}^n)\phi_{cl}^{(\mathcal{T})} + \tilde{\tau}^n\hat{\phi}_{cl}$, where $n$ represents the number of iterations that have not been updated since the last parameter update for $\phi_{cl}$. We also implement GMMs to estimate the anchor distributions $P_p$ and $P_n$ using the source domain data's global positive and negative logits. The algorithm flow is detailed in Appendix D.

After estimating the logits distributions using GMMs, we compute the adjusted logits offset $\Delta^S$ and $\Delta^T$ for source and target domains, respectively. These offsets are then used to adjust the cross-entropy loss in Eq.1 and 2 for both domains to abtain $\widetilde{\mathcal{L}}^s$ and $\widetilde{\mathcal{L}}^u$ :

$$\widetilde{\ell}_{ce}^s = -\log \frac{\exp(f_\theta(x_{ij}^S)[y_{ij}^S] - \tau\Delta_{y_{ij}^S, y_{ij}^S}^S(f_\theta(x_{ij}^S)[y_{ij}^S]))}{\sum_{c=1}^{C} \exp(f_\theta(x_{ij}^S)[c] - \tau\Delta_{y_{ij}^S, c}^S(f_\theta(x_{ij}^S)[c]))},$$

$$\widetilde{\ell}_{ce}^u = -\log \frac{\exp(f_\theta(x_{ij}^T)[\hat{y}_{ij}^T] - \tau\Delta_{\hat{y}_{ij}^T, \hat{y}_{ij}^T}^T(f_\theta(x_{ij}^T)[\hat{y}_{ij}^T]))}{\sum_{c=1}^{C} \exp(f_\theta(x_{ij}^T)[c] - \tau\Delta_{\hat{y}_{ij}^T, c}^T(f_\theta(x_{ij}^T)[c]))}.$$
$$(9)$$

In contrast to Eq.7, where the logits are adjusted post-hoc, Eq.9 directly incorporates the offset into the learning process of the logits. This approach is equivalent to learning a scorer of the form $g(x)[y] = f(x)[y] - \tau\Delta_y(x)$. Consequently, we have $\arg\max f(x)[y] = g(x)[y] + \tau\Delta_y(x)$, which can be seen as analogous to the post-hoc adjustment. In Appendix B, we also demonstrate that Eq.9 can be seen as an adaptive margin-based loss. By employing these adjusted losses, we achieve a two-fold benefit: (1) The anchor distribution can serve as a reference distribution to balance

the learning progress between classes within both domains. (2) Since the pseudo-label-based loss in the target domain has a gradually increasing weight, using a shared anchor distribution allows the logits distribution of the target domain to gradually align with that of the source domain, thus establishing a connection between the two domains.

### 3.3.5. BRIDGING DOMAINS THROUGH CUMULATIVE DENSITY ESTIMATION

Furthermore, for each sample pixel, we can query the corresponding positive cumulative distribution value $F_{cc}$ based on its label $c$, which ranges from 0 to 1. The positive distribution measures the discriminative ability of a class, and we find that this cumulative distribution value indicates the difficulty of the sample pixel belonging to that class. This structural knowledge depends only on the context of the pixel and is not affected by the image style, making it domain-invariant. To further bridge the two domains, we add an extra regression head to the network to predict this value as an additional auxiliary task.

Specifically, for each sample point $x_{ij}^S$ in the source domain, we can query the corresponding positive cumulative distribution value based on its true label $y_{ij}^S$: $d_{ij}^S = F_{y_{ij}^S, y_{ij}^S}(f_\theta(x_{ij}^S)[y_{ij}^S])$, where $F_{y_{ij}^S, y_{ij}^S}$ is the positive cumulative distribution function for class $y_{ij}^S$. Similarly, for each sample point $x_{ij}^T$ in the target domain, we can query the corresponding positive cumulative distribution value based on its pseudo label $\hat{y}_{ij}^T$: $d_{ij}^T = F_{\hat{y}_{ij}^T, \hat{y}_{ij}^T}(f_\theta(x_{ij}^T)[\hat{y}_{ij}^T])$, where $F_{\hat{y}_{ij}^T, \hat{y}_{ij}^T}$ is the positive cumulative distribution function for the class corresponding to the pseudo label $\hat{y}_{ij}^T$. We then add an extra regression head $h_\phi$ to the network to predict the cumulative distribution value for each sample point. The corresponding regression losses for the source and target domains can be defined as:

$$\mathcal{L}_{reg}^S = \frac{1}{N_S} \sum_{i=1}^{N_S} \sum_{j=1}^{H \times W} |h_\phi(\tilde{f}_\theta(x_{ij}^S)) - d_{ij}^S|^2,$$

$$\mathcal{L}_{reg}^T = \frac{1}{N_T} \sum_{i=1}^{N_T} \sum_{j=1}^{H \times W} q(p_{ij})|h_\phi(\tilde{f}_\theta(x_{ij}^T)) - d_{ij}^T|^2,$$

(10)

where $|\cdot|^2$ denotes the L2 loss, and $\tilde{f}_\theta(x_{ij})$ denote the features extracted by the network $f_\theta$. Finally, the overall training objective can be expressed as $\mathcal{L} = \widetilde{\mathcal{L}}^s + \widetilde{\mathcal{L}}^u + \lambda(\mathcal{L}_{reg}^S + \mathcal{L}_{reg}^T)$, where $\lambda$ is a hyperparameter balancing the cumulative density estimation loss.

## 4. Experiments

### 4.1. Experimental Setup

**Datasets.** Following standard UDA protocols, we evaluate our method on two widely used benchmarks that in-

volve transferring knowledge from a synthetic domain to a real domain in a street scene setting. Specifically, we use GTAv/SYNTHIA (Ros et al., 2016; Richter et al., 2016) as the labeled source domain and Cityscapes (Cordts et al., 2016) as the unlabeled target domain. GTAv contains 24,966 synthetic images with a resolution of $1914 \times 1052$, while SYNTHIA consists of 9,400 synthetic images with a resolution of $1280 \times 960$.

**Implementation Details.** Our method can be built with different self-training-based frameworks. For thorough evaluation, we apply BLDA to four strong baseline methods, i.e., DAFormer (Hoyer et al., 2022a), CDAC (Wang et al., 2023), HRDA (Hoyer et al., 2022b), and MIC (Hoyer et al., 2023), with MiT-B5 (Xie et al., 2021) pretrained on ImageNet-1k (Deng et al., 2009) as the backbone. We also implement CNN-based (He et al., 2016) backbone with DACS (Tranheden et al., 2021), and DAFormer(C). BLDA is implemented based on MMSegmentation (Contributors, 2020). All experiments are trained for 40K iterations and a batch size of 2, with one or two RTX-3090 (24 GB memory) GPUs, depending on the complexity of used UDA frameworks. We train the network with an AdamW optimizer with learning rates of $6 \times 10^{-5}$ for the encoder and $6 \times 10^{-4}$ for the decoder, a weight decay of 0.01, and linear learning rate warm-up for the first 1.5K iterations. The input images are rescaled and randomly cropped to $512 \times 512$ following the same data augmentation in DAFormer (Hoyer et al., 2022a), and the EMA coefficient for updating the teacher net is set to be 0.999. We set temperature coefficient $\tau = 0.1$ and loss weight $\lambda = 0.2$ respectively.

### 4.2. Comparison with Existing Methods

**Comparative Evaluation.** We compare BLDA to existing state-of-the-art UDA approaches on the GTA.→CS. and SYN.→CS. benchmarks. In addition to the widely adopted mean intersection-over-union (mIoU) metric, we also report the mean accuracy (mAcc) metric, which is equivalent to measuring the balanced error (Menon et al., 2020), i.e., the average of each class's error rate, and is more suitable to assess the balance among classes. We discuss these two metrics in detail in Appendix C. Additionally, we calculate the standard deviation of IoU and Acc for each class to reflect the balanced degree of performance across classes.

**Evaluation Results.** Tab.1-2 shows BLDA consistently improves the performance of all baseline methods on two benchmarks by a large margin, ranging from 1.2% to 3.1%. Furthermore, significant improvements are obtained for under-predicted classes, such as *sidewalk*, *fence*, *pole*, *light*, and *sign*, which demonstrates that BLDA can mitigate the class bias with decreased standard deviation and thus bring the performance gains. In Table 3, the same phenomenon is observed, and the improvement in mAcc is more significant,

Table 1: UDA segmentation performance on GTA.→CS. using the mIoU (%) evaluation metric, where the improvement is marked as **bold**. The results are acquired based on CNN-based model (He et al., 2016; Chen et al., 2017a), denoted as C, and Transformer-based model (Xie et al., 2021), denoted as T. ∗ denotes the reproduced result.

| Method | Arch. | Road | Sidewalk | Building | Wall | Fence | Pole | Light | Sign | Veg | Terrain | Sky | Person | Rider | Car | Truck | Bus | Train | Motor | Bike | mIoU (↑) | std (↓) |
|---|---|---|---|---|---|---|---|---|---|---|---|---|---|---|---|---|---|---|---|---|---|---|
| AdaptSegNet (Tsai et al., 2018) | C | 86.5 | 36.0 | 79.9 | 23.4 | 23.3 | 23.9 | 35.2 | 14.8 | 83.4 | 33.3 | 75.6 | 58.5 | 27.6 | 73.7 | 32.5 | 35.4 | 3.9 | 30.1 | 28.1 | 42.4 | 24.7 |
| CRST (Zou et al., 2019) | C | 91.7 | 45.1 | 80.9 | 29.0 | 23.4 | 43.8 | 47.1 | 40.9 | 84.0 | 20.0 | 60.6 | 64.0 | 31.9 | 85.8 | 39.5 | 48.7 | 25.0 | 38.0 | 47.0 | 49.8 | 21.6 |
| PLCA (Kang et al., 2020) | C | 84.0 | 30.4 | 82.4 | 35.3 | 24.8 | 32.2 | 36.8 | 24.5 | 85.5 | 37.2 | 78.6 | 66.9 | 32.8 | 85.5 | 40.4 | 48.0 | 8.8 | 29.8 | 41.8 | 47.7 | 23.9 |
| ProDA (Zhang et al., 2021) | C | 87.8 | 56.0 | 79.7 | 46.3 | 44.8 | 45.6 | 53.5 | 53.5 | 88.6 | 45.2 | 82.1 | 70.7 | 39.2 | 88.8 | 45.5 | 50.4 | 1.0 | 48.9 | 56.4 | 57.5 | 21.2 |
| CPSL (Li et al., 2022) | C | 92.3 | 59.5 | 84.9 | 45.7 | 29.7 | 52.8 | 61.5 | 59.5 | 87.9 | 41.6 | 85.0 | 73.0 | 35.5 | 90.4 | 48.7 | 73.9 | 26.3 | 53.8 | 53.9 | 60.8 | 20.3 |
| TransDA (Chen et al., 2022) | T | 94.7 | 64.2 | 89.2 | 48.1 | 45.8 | 50.1 | 60.2 | 40.8 | 90.4 | 50.2 | 93.7 | 76.7 | 47.6 | 92.5 | 56.8 | 60.1 | 47.6 | 49.6 | 55.4 | 63.9 | 18.5 |
| ADFormer (He & Todorovic, 2025) | T | 96.7 | 75.1 | 88.8 | 57.5 | 45.9 | 45.6 | 55.4 | 59.8 | 90.2 | 45.6 | 92.1 | 70.8 | 43.0 | 91.0 | 78.9 | 79.3 | 68.7 | 52.7 | 65.0 | 69.2 | 17.6 |
| DIGA (Shen et al., 2023) | T | 97.0 | 78.6 | 91.3 | 60.8 | 56.7 | 56.5 | 64.4 | 69.9 | 91.5 | 50.8 | 93.7 | 79.2 | 55.2 | 93.7 | 78.3 | 86.9 | 77.8 | 63.7 | 65.8 | 74.3 | 14.7 |
| CoPT (Mata et al., 2025) | T | 97.6 | 80.9 | 91.6 | 62.1 | 55.9 | 59.3 | 66.7 | 70.5 | 91.9 | 53.0 | 94.4 | 80.0 | 55.6 | 94.7 | 87.1 | 88.6 | 82.1 | 65.0 | 68.8 | 76.1 | 14.7 |
| DACS∗ (Tranheden et al., 2021) | C | 93.0 | 52.0 | 87.8 | 29.4 | 38.3 | 37.7 | 45.0 | 53.3 | 87.9 | 46.3 | 90.2 | 67.8 | 38.0 | 89.0 | 51.1 | 51.1 | 0.0 | 10.7 | 19.4 | 52.1 | 27.1 |
| +BLDA | C | 92.9 | **67.5** | 87.1 | **36.3** | **39.3** | **41.2** | **50.7** | **58.5** | 87.3 | 45.5 | 87.7 | **69.1** | **40.4** | 88.3 | 45.3 | **53.5** | 1.2 | **10.8** | **36.3** | **54.7** | 25.6 |
| DAFormer(C)∗ (Hoyer et al., 2022a) | C | 95.7 | 69.9 | 87.2 | 35.6 | 36.7 | 37.0 | 49.4 | 52.8 | 87.3 | 44.1 | 87.9 | 69.0 | 42.2 | 86.5 | 40.0 | 51.7 | 0.2 | 41.1 | 54.0 | 56.2 | 24.0 |
| +BLDA | C | 95.6 | **73.6** | 86.7 | **41.0** | **40.2** | **43.3** | **51.1** | **62.4** | 86.2 | 43.7 | 87.4 | 68.3 | 39.2 | **86.6** | **43.6** | 45.6 | 1.0 | **49.4** | **59.4** | **58.1** | 23.2 |
| DAFormer (Hoyer et al., 2022a) | T | 95.7 | 70.2 | 89.4 | 53.5 | 48.1 | 49.6 | 55.8 | 59.4 | 89.9 | 47.9 | 92.5 | 72.2 | 44.7 | 92.3 | 74.5 | 78.2 | 65.1 | 55.9 | 61.8 | 68.3 | 16.8 |
| +BLDA | T | 95.4 | **78.3** | 88.3 | **54.0** | **55.2** | **55.7** | **60.3** | **65.2** | 89.2 | 47.3 | 91.1 | 71.4 | **44.8** | 91.6 | 74.3 | **83.4** | **73.2** | **59.3** | **67.1** | **70.7** | 15.5 |
| CDAC∗ (Wang et al., 2023) | T | 96.1 | 72.8 | 90.5 | 55.2 | 48.0 | 51.8 | 57.1 | 61.8 | 90.8 | 50.4 | 91.9 | 73.2 | 46.9 | 93.6 | 80.7 | 78.6 | 58.2 | 56.9 | 64.5 | 69.2 | 16.7 |
| +BLDA | T | **96.6** | **78.1** | **90.0** | **57.9** | **52.5** | **55.1** | **58.7** | **64.5** | 90.1 | **50.8** | 90.9 | **73.3** | **47.5** | 93.2 | 75.1 | **80.0** | **65.3** | 60.7 | **68.9** | **71.0** | 15.4 |
| HRDA (Hoyer et al., 2022b) | T | 96.5 | 74.4 | 91.0 | 61.6 | 51.5 | 57.1 | 63.9 | 69.3 | 91.3 | 48.4 | 94.2 | 79.0 | 52.9 | 93.9 | 84.1 | 85.7 | 75.9 | 63.9 | 67.5 | 73.8 | 15.4 |
| +BLDA | T | 96.4 | **77.6** | 90.7 | **63.3** | **57.9** | **62.1** | **66.5** | **72.5** | 91.3 | **52.2** | **94.4** | 76.9 | **57.3** | 93.5 | **86.2** | **87.7** | **79.9** | **66.8** | **68.9** | **75.6** | 13.8 |
| MIC (Hoyer et al., 2023) | T | 97.4 | 80.1 | 91.7 | 61.2 | 56.9 | 59.7 | 66.0 | 71.3 | 91.7 | 51.4 | 94.3 | 79.8 | 56.1 | 94.6 | 85.4 | 90.3 | 80.4 | 64.5 | 68.5 | 75.9 | 14.8 |
| +BLDA | T | 97.1 | **82.6** | 91.6 | **64.7** | **61.0** | **64.9** | **68.0** | **74.8** | 91.2 | **56.6** | 92.4 | **80.0** | 54.7 | **95.7** | **87.3** | 88.8 | **82.6** | 64.2 | **72.0** | **77.1** | 13.5 |

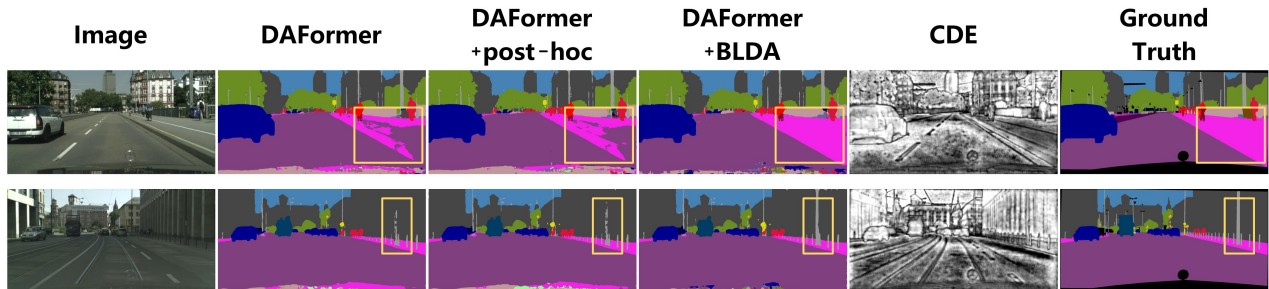

Figure 4: Qualitative results. Note that the yellow boxes mark regions improved by BLDA.

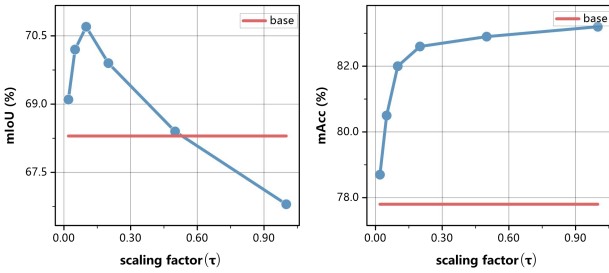

Figure 5: Study of the different evalution metrics with respect to scaling factor $\tau$.

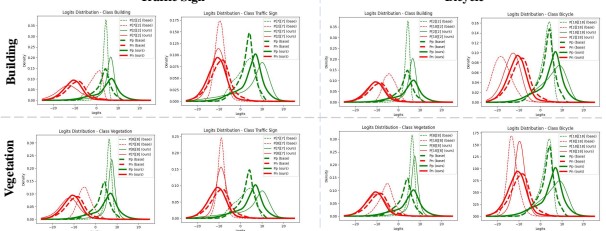

Figure 6: Comparison of Logits Distribution. We choose {*building* (2), *vegetation* (8)} as over-predicted classes, and {*traffic sign* (7), *bicycle* (18)} as under-predicted classes for visualization. Note that the anchor distribution is counted separately at baseline and in our method.

ranging from 2.9% to 5.6%.

### 4.3. Ablation Study

We conduct a series of ablation studies on the GTA.→CS. benchmark built with DAFormer (Hoyer et al., 2022a) in this section. Please refer to the Appendix E-O for further analysis, where we provide more experiment results, deeper ablation studies, and more visualization.

**Influence of scaling factor.** As illustrated in Fig. 5, we explore the influence of different $\tau$ values on evaluation metrics. The mAcc metric gradually increases as $\tau$ grows, reaching its peak performance when $\tau = 1$. Interestingly, the mIoU metric does not demonstrate a perfectly positive correlation with the rise in mAcc. This discrepancy arises

Table 2: UDA segmentation performance on SYN.→CS. using the mIoU (%) evaluation metric, where the improvement is marked as **bold**. Note that the mIoUs on are calculated over 16 classes.

| Method | Arch. | Road | Sidewalk | Building | Wall | Fence | Pole | Light | Sign | Veg | Terrain | Sky | Person | Rider | Car | Truck | Bus | Train | Motor | Bike | mIoU (↑) | std (↓) |
|---|---|---|---|---|---|---|---|---|---|---|---|---|---|---|---|---|---|---|---|---|---|---|
| CRST (Zou et al., 2019) | C | 67.7 | 32.2 | 73.9 | 10.7 | 1.6 | 37.4 | 22.2 | 31.2 | 80.8 | - | 80.5 | 60.8 | 29.1 | 82.8 | - | 25.0 | - | 19.4 | 45.3 | 43.8 | 26.0 |
| PLCA (Kang et al., 2020) | C | 82.6 | 29.0 | 81.0 | 11.2 | 0.2 | 33.6 | 24.9 | 18.3 | 82.8 | - | 82.3 | 62.1 | 26.5 | 85.6 | - | 48.9 | - | 26.8 | 52.2 | 46.8 | 28.3 |
| DACS (Tranheden et al., 2021) | C | 80.6 | 25.1 | 81.9 | 21.5 | 2.9 | 37.2 | 22.7 | 24.0 | 83.7 | - | 90.8 | 67.6 | 38.3 | 82.9 | - | 38.9 | - | 28.5 | 47.6 | 48.3 | 27.5 |
| ProDA (Zhang et al., 2021) | C | 87.8 | 45.7 | 84.6 | 37.1 | 0.6 | 44.0 | 54.6 | 37.0 | 88.1 | - | 84.4 | 74.2 | 24.3 | 88.2 | - | 51.1 | - | 40.5 | 45.6 | 55.5 | 25.6 |
| CPSL (Li et al., 2022) | C | 87.2 | 43.9 | 85.5 | 33.6 | 0.3 | 47.7 | 57.4 | 37.2 | 87.8 | - | 88.5 | 79.0 | 32.0 | 90.6 | - | 49.4 | - | 50.8 | 59.8 | 57.9 | 25.5 |
| TransDA (Chen et al., 2022) | T | 90.4 | 54.8 | 86.4 | 31.1 | 1.7 | 53.8 | 61.1 | 37.1 | 90.3 | - | 93.0 | 71.2 | 25.3 | 92.3 | - | 66.0 | - | 44.4 | 49.8 | 59.3 | 26.5 |
| ADFormer (He & Todorovic, 2025) | T | 91.8 | 53.6 | 87.0 | 40.5 | 5.2 | 46.8 | 52.1 | 54.9 | 88.4 | - | 92.6 | 72.5 | 45.7 | 86.1 | - | 61.6 | - | 50.4 | 64.4 | 62.1 | 22.9 |
| DIGA (Shen et al., 2023) | T | 85.2 | 47.7 | 88.8 | 49.5 | 4.8 | 57.2 | 65.7 | 60.9 | 85.3 | - | 92.9 | 79.4 | 52.8 | 89.0 | - | 64.7 | - | 63.9 | 64.9 | 65.8 | 21.4 |
| CoPT (Mata et al., 2025) | T | 83.4 | 44.3 | 90.0 | 50.4 | 8.0 | 60.0 | 67.0 | 63.0 | 87.5 | - | 94.8 | 81.1 | 58.6 | 89.7 | - | 66.5 | - | 68.9 | 65.0 | 67.4 | 21.1 |
| DAFormer (Hoyer et al., 2022a) | T | 84.5 | 40.7 | 88.4 | 41.5 | 6.5 | 50.0 | 55.0 | 54.6 | 86.0 | - | 89.8 | 73.2 | 48.2 | 87.2 | - | 53.2 | - | 53.9 | 61.7 | 60.9 | 22.1 |
| **+BLDA** | T | 80.7 | **44.9** | 85.6 | **45.1** | **9.6** | **54.3** | **60.2** | **58.7** | **87.7** | - | **92.3** | **75.7** | **51.1** | **87.3** | - | **62.7** | - | **59.9** | **65.8** | **64.0** | **20.6** |
| HRDA (Hoyer et al., 2022b) | T | 85.2 | 47.7 | 88.8 | 49.5 | 4.8 | 57.2 | 65.7 | 60.9 | 85.3 | - | 92.9 | 79.4 | 52.8 | 89.0 | - | 64.7 | - | 63.9 | 64.9 | 65.8 | 21.4 |
| **+BLDA** | T | 83.9 | **54.9** | 87.5 | **53.1** | **11.5** | **63.2** | **69.4** | **64.4** | **87.2** | - | **93.1** | 79.1 | **54.7** | 88.3 | - | **69.1** | - | **64.2** | **65.7** | **67.9** | **19.3** |
| MIC (Hoyer et al., 2023) | T | 86.6 | 50.5 | 89.3 | 47.9 | 7.8 | 59.4 | 66.7 | 63.4 | 87.1 | - | 94.6 | 81.0 | 58.9 | 90.1 | - | 61.9 | - | 67.1 | 64.3 | 67.3 | 21.0 |
| **+BLDA** | T | 86.1 | **61.2** | **89.8** | 47.2 | **10.2** | **62.6** | **70.3** | **67.1** | **90.0** | - | 94.4 | **81.4** | 56.4 | **90.5** | - | **67.2** | - | 64.8 | **66.3** | **69.1** | **20.4** |

Table 3: UDA segmentation performance on GTA.→CS. using the mAcc (%) evaluation metric, where the improvement is marked as **bold**. ∗ denotes the reproduced result.

| Method | Arch. | Road | Sidewalk | Building | Wall | Fence | Pole | Light | Sign | Veg | Terrain | Sky | Person | Rider | Car | Truck | Bus | Train | Motor | Bike | mAcc (↑) | std (↓) |
|---|---|---|---|---|---|---|---|---|---|---|---|---|---|---|---|---|---|---|---|---|---|---|
| DACS∗ (Tranheden et al., 2021) | C | 97.9 | 58.9 | 94.9 | 40.7 | 49.4 | 46.1 | 51.7 | 57.2 | 94.5 | 66.9 | 98.6 | 82.0 | 62.2 | 93.3 | 82.9 | 79.0 | 0.0 | 74.4 | 20.0 | 65.8 | 26.5 |
| **+BLDA** | C | 97.1 | **80.5** | 93.4 | **54.1** | 39.2 | **50.5** | **64.6** | **67.4** | 93.7 | **71.5** | **99.0** | **83.4** | 58.8 | **95.0** | 73.0 | 76.2 | **1.0** | 73.5 | **39.2** | **69.0** | **24.2** |
| DAFormer(C)∗ (Hoyer et al., 2022a) | C | 98.4 | 78.3 | 93.6 | 43.2 | 45.4 | 45.0 | 61.5 | 59.5 | 93.8 | 65.3 | 98.0 | 80.1 | 69.2 | 92.2 | 77.0 | 85.3 | 0.2 | 69.7 | 60.8 | 69.3 | 23.8 |
| **+BLDA** | C | 97.0 | **84.8** | 93.0 | **55.1** | **56.1** | **55.4** | **71.0** | **74.5** | 93.2 | **73.7** | 97.7 | **84.8** | **77.2** | 92.0 | **80.1** | **85.5** | **1.1** | **77.3** | **73.0** | **74.9** | **21.6** |
| DAFormer (Hoyer et al., 2022a) | T | 99.1 | 74.8 | 95.2 | 61.0 | 53.1 | 59.8 | 70.7 | 68.6 | 96.0 | 63.4 | 98.7 | 84.2 | 69.9 | 95.5 | 88.9 | 84.1 | 74.0 | 70.0 | 69.7 | 77.8 | 14.2 |
| **+BLDA** | T | 98.3 | **86.7** | 93.4 | **70.9** | **61.9** | **66.0** | **74.0** | **78.9** | 95.4 | 59.0 | 98.6 | **85.7** | **73.1** | 95.3 | **89.8** | **89.3** | **85.4** | **77.2** | **78.6** | **82.0** | **11.9** |
| CDAC∗ (Wang et al., 2023) | T | 99.1 | 78.0 | 94.6 | 69.3 | 52.9 | 61.2 | 71.7 | 68.7 | 95.8 | 59.3 | 99.0 | 85.0 | 70.2 | 96.0 | 88.3 | 85.5 | 71.1 | 75.4 | 74.7 | 78.7 | 13.8 |
| **+BLDA** | T | **98.1** | 82.5 | **94.0** | 74.8 | 66.7 | 68.0 | 76.8 | 74.1 | 95.5 | 67.9 | 98.3 | 87.1 | 72.9 | 95.6 | 90.8 | 88.8 | 71.8 | 80.8 | 82.9 | 82.5 | 11.5 |
| HRDA (Hoyer et al., 2022b) | T | 99.1 | 85.6 | 96.0 | 72.5 | 56.3 | 69.4 | 83.9 | 79.6 | 96.1 | 59.1 | 98.5 | 89.8 | 60.7 | 96.2 | 91.3 | 89.4 | 80.2 | 77.0 | 80.9 | 82.2 | 13.2 |
| **+BLDA** | T | **99.2** | **87.2** | 95.0 | 64.9 | **68.2** | **72.7** | **88.3** | 79.5 | 95.7 | **65.7** | **98.9** | 87.9 | **79.6** | 95.5 | **93.8** | **92.5** | **87.0** | **83.1** | **81.2** | **85.1** | **10.7** |
| MIC (Hoyer et al., 2023) | T | 99.5 | 87.1 | 96.0 | 73.2 | 65.3 | 68.5 | 81.0 | 74.8 | 96.8 | 58.9 | 98.8 | 85.7 | 81.3 | 96.7 | 91.4 | 91.0 | 78.1 | 79.0 | 77.5 | 83.2 | 11.6 |
| **+BLDA** | T | 98.9 | **90.7** | 95.4 | **74.6** | **70.9** | **73.5** | **88.9** | **82.8** | 96.4 | **64.8** | 98.8 | **89.0** | 80.8 | 96.3 | **93.8** | **92.8** | **88.2** | **86.5** | **78.8** | **86.3** | **9.8** |

Table 4: BLDA ablation study of different components.

| None | Post-hoc | OLA$_S$ | OLA | CDE | mIoU | mAcc |
|---|---|---|---|---|---|---|
| ✓ | | | | | 68.3 | 77.8 |
| | ✓ | | | | 69.2 | 80.3 |
| | | ✓ | | | 68.9 | 79.5 |
| | | | ✓ | | 70.2 | 81.8 |
| | | | ✓ | ✓ | **70.7** | **82.0** |

from the fact that the mIoU calculation is heavily impacted by the imbalanced distribution of the test set, whereas mAcc serves as a class-balanced metric. We comprehensively explain this phenomenon in Appendix C. Our method, which models class-balanced learning, effectively boosts mAcc. However, to achieve improvements in mIoU, selecting a smaller scaling factor $\tau$ is necessary.

**Effectiveness of Components.** In Tab.4, we delve into the various components of BLDA. By solely applying the post-hoc method to adjust the predictions, we observe a minor performance improvement of 0.9%. When introducing online logit adjustment exclusively during source domain image training (OLA$_S$), the improvement is comparatively modest at 0.6%. However, by simultaneously performing adjustments in both domains (OLA), we witness a significant

performance boost of 1.9%, suggesting that this strategy effectively captures the disparity in learning degrees between the domains. Lastly, the extra supervison from cumulative distribution estimation (CDE) further models the shared structural information across the domains (shown in Fig.4), producing additional performance gains.

**Qualitative Results.** Fig.4 presents the qualitative results. We observe that for under-predicted classes, such as *sidewalk* and *pole*, the baseline method struggles to recognize them accurately. While the post-hoc method can slightly improve the performance, our proposed BLDA approach significantly enhances the ability to predict these classes.

**Comparison of Logits Distribution.** In Fig.6, we visualize the positive distribution and negative distribution corresponding to the over-predicted classes {*building* (2), *vegetation* (8)} and under-predicted classes {*traffic sign* (7), *bicycle* (18)} on the Cityscapes Val. set. In the baseline method, the positive and negative logits of classes *building* and *vegetation* are larger than anchor distribution, while this phenomenon is reversed in classes *traffic light* and *bicycle*, which leads to class bias. Our method reduces this distribution difference by aligning with the anchor distribution and achieves class-balanced learning.

# 5. Conclusion

In this work, we present Balanced Learning for Domain Adaptation (BLDA), a novel approach to address class bias in unsupervised domain adaptation (UDA) for semantic segmentation. BLDA analyzes logits distributions to assess prediction bias and introduces an online logits adjustment mechanism to balance class learning in both source and target domains. Our method effectively mitigates class bias, promotes balanced learning, and enhances generalization to the target domain. Experimental results demonstrate consistent performance improvements when integrating BLDA with various methods on standard UDA benchmarks. The extensive experiments across different tasks, baselines, and architectures showcase the effectiveness and versatility of BLDA in addressing class bias in UDA settings.

# Acknowledgements

This work was partially supported by the Open Fund of National Key Laboratory of Deep Space Exploration (Grant NKLDSE2023A009).

# Impact Statement

Within this paper, we present an approach for domain adaptive semantic segmentation, a pivotal research area in the realm of computer vision, with no apparent negative societal implications known thus far.

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

Contributors, M. Mmsegmentation: Openmmlab semantic segmentation toolbox and benchmark, 2020.

Cordts, M., Omran, M., Ramos, S., Rehfeld, T., Enzweiler, M., Benenson, R., Franke, U., Roth, S., and Schiele, B. The cityscapes dataset for semantic urban scene understanding. In *Proceedings of the IEEE conference on computer vision and pattern recognition*, pp. 3213–3223, 2016.

Cui, Y., Jia, M., Lin, T.-Y., Song, Y., and Belongie, S. Class-balanced loss based on effective number of samples. In *Proceedings of the IEEE/CVF conference on computer vision and pattern recognition*, pp. 9268–9277, 2019.

Deng, J., Dong, W., Socher, R., Li, L.-J., Li, K., and Fei-Fei, L. Imagenet: A large-scale hierarchical image database. In *2009 IEEE conference on computer vision and pattern recognition*, pp. 248–255. Ieee, 2009.

Ganin, Y. and Lempitsky, V. Unsupervised domain adaptation by backpropagation. In *International conference on machine learning*, pp. 1180–1189. PMLR, 2015.

Guan, D., Huang, J., Xiao, A., and Lu, S. Unbiased subclass regularization for semi-supervised semantic segmentation. In *Proceedings of the IEEE/CVF Conference on Computer Vision and Pattern Recognition*, pp. 9968–9978, 2022.

Guo, C., Pleiss, G., Sun, Y., and Weinberger, K. Q. On calibration of modern neural networks. In *International conference on machine learning*, pp. 1321–1330. PMLR, 2017.

He, H. and Garcia, E. A. Learning from imbalanced data. *IEEE Transactions on knowledge and data engineering*, 21(9):1263–1284, 2009.

He, H., Bai, Y., Garcia, E. A., and Li, S. Adasyn: Adaptive synthetic sampling approach for imbalanced learning. In *2008 IEEE international joint conference on neural networks (IEEE world congress on computational intelligence)*, pp. 1322–1328. Ieee, 2008.

He, K., Zhang, X., Ren, S., and Sun, J. Deep residual learning for image recognition. In *Proceedings of the IEEE conference on computer vision and pattern recognition*, pp. 770–778, 2016.

He, L. and Todorovic, S. Attention decomposition for cross-domain semantic segmentation. In *European Conference on Computer Vision*, pp. 414–431. Springer, 2025.

He, R., Yang, J., and Qi, X. Re-distributing biased pseudo labels for semi-supervised semantic segmentation: A baseline investigation. In *Proceedings of the IEEE/CVF International Conference on Computer Vision*, pp. 6930–6940, 2021.

Hong, W., Wang, Z., Yang, M., and Yuan, J. Conditional generative adversarial network for structured domain adaptation. In *Proceedings of the IEEE conference on computer vision and pattern recognition*, pp. 1335–1344, 2018.

Hoyer, L., Dai, D., and Van Gool, L. Daformer: Improving network architectures and training strategies for domain-adaptive semantic segmentation. In *Proceedings of the IEEE/CVF Conference on Computer Vision and Pattern Recognition*, pp. 9924–9935, 2022a.

Hoyer, L., Dai, D., and Van Gool, L. Hrda: Context-aware high-resolution domain-adaptive semantic segmentation. In *European Conference on Computer Vision*, pp. 372–391. Springer, 2022b.

Hoyer, L., Dai, D., Wang, H., and Van Gool, L. Mic: Masked image consistency for context-enhanced domain adaptation. In *Proceedings of the IEEE/CVF Conference on Computer Vision and Pattern Recognition*, pp. 11721–11732, 2023.

Huo, X., Xie, L., Zhou, W., Li, H., and Tian, Q. Focus on your target: A dual teacher-student framework for domain-adaptive semantic segmentation. *arXiv preprint arXiv:2303.09083*, 2023.

Jin, Y., Wang, X., Long, M., and Wang, J. Minimum class confusion for versatile domain adaptation. In *Computer Vision–ECCV 2020: 16th European Conference, Glasgow, UK, August 23–28, 2020, Proceedings, Part XXI 16*, pp. 464–480. Springer, 2020.

Kang, G., Wei, Y., Yang, Y., Zhuang, Y., and Hauptmann, A. Pixel-level cycle association: A new perspective for domain adaptive semantic segmentation. *Advances in neural information processing systems*, 33:3569–3580, 2020.

Kareer, S., Vijaykumar, V., Maheshwari, H., Chattopadhyay, P., Hoffman, J., and Prabhu, V. We're not using videos effectively: An updated domain adaptive video segmentation baseline. *arXiv preprint arXiv:2402.00868*, 2024.

Kim, J., Jeong, J., and Shin, J. M2m: Imbalanced classification via major-to-minor translation. In *Proceedings of the IEEE/CVF conference on computer vision and pattern recognition*, pp. 13896–13905, 2020.

Li, R., Li, S., He, C., Zhang, Y., Jia, X., and Zhang, L. Class-balanced pixel-level self-labeling for domain adaptive semantic segmentation. In *Proceedings of the IEEE/CVF Conference on Computer Vision and Pattern Recognition*, pp. 11593–11603, 2022.

Li, W., Sun, R., and Zhang, T. A universal degradation-based bridging technique for domain adaptive semantic segmentation. *arXiv preprint arXiv:2412.10339*, 2024.

Li, Z., Li, W., Mai, H., Zhang, T., and Xiong, Z. Enhancing cell detection in histopathology images: a vit-based u-net approach. In *International Conference on Medical Image Computing and Computer-Assisted Intervention*, pp. 150–160. Springer, 2023.

Li, Z., Wang, Y., Li, W., Zhang, T., and Liu, X. Dual-agent optimization framework for cross-domain few-shot segmentation. In *Proceedings of the IEEE/CVF Conference on Computer Vision and Pattern Recognition*, 2025.

Lin, T.-Y., Goyal, P., Girshick, R., He, K., and Dollár, P. Focal loss for dense object detection. In *Proceedings of the IEEE international conference on computer vision*, pp. 2980–2988, 2017.

Liu, Z., Miao, Z., Zhan, X., Wang, J., Gong, B., and Yu, S. X. Large-scale long-tailed recognition in an open world. In *Proceedings of the IEEE/CVF conference on computer vision and pattern recognition*, pp. 2537–2546, 2019.

Long, J., Shelhamer, E., and Darrell, T. Fully convolutional networks for semantic segmentation. In *Proceedings of the IEEE conference on computer vision and pattern recognition*, pp. 3431–3440, 2015a.

Long, M., Cao, Y., Wang, J., and Jordan, M. Learning transferable features with deep adaptation networks. In *International conference on machine learning*, pp. 97–105. PMLR, 2015b.

Luo, N., Wang, Y., Sun, R., Xiong, G., Zhang, T., and Wu, F. Exploring the better correlation for few-shot video object segmentation. *IEEE Transactions on Circuits and Systems for Video Technology*, 2024.

Mai, H., Sun, R., Wang, Y., Zhang, T., and Wu, F. Pay attention to target: Relation-aware temporal consistency for domain adaptive video semantic segmentation. In *Proceedings of the AAAI Conference on Artificial Intelligence*, volume 38, pp. 4162–4170, 2024a.

Mai, H., Sun, R., Zhang, T., and Wu, F. Rankmatch: Exploring the better consistency regularization for semi-supervised semantic segmentation. In *Proceedings of the IEEE/CVF Conference on Computer Vision and Pattern Recognition*, pp. 3391–3401, 2024b.

Mata, C., Ranasinghe, K., and Ryoo, M. S. Copt: Unsupervised domain adaptive segmentation using domain-agnostic text embeddings. In *European Conference on Computer Vision*, pp. 424–440. Springer, 2025.

McLachlan, G. J. and Krishnan, T. *The EM algorithm and extensions*. John Wiley & Sons, 2007.

Menon, A. K., Jayasumana, S., Rawat, A. S., Jain, H., Veit, A., and Kumar, S. Long-tail learning via logit adjustment. *arXiv preprint arXiv:2007.07314*, 2020.

Moradinasab, N., Shankman, L. S., Deaton, R. A., Owens, G. K., and Brown, D. E. Protogmm: Multi-prototype gaussian-mixture-based domain adaptation model for semantic segmentation. *arXiv preprint arXiv:2406.19225*, 2024.

Ning, M., Lu, D., Wei, D., Bian, C., Yuan, C., Yu, S., Ma, K., and Zheng, Y. Multi-anchor active domain adaptation for semantic segmentation. In *Proceedings of the IEEE/CVF International Conference on Computer Vision*, pp. 9112–9122, 2021.

Pan, Y., Sun, R., Luo, N., Zhang, T., and Zhang, Y. Exploring reliable matching with phase enhancement for night-time semantic segmentation. In *European Conference on Computer Vision*, pp. 408–424. Springer, 2024.

Peng, D., Hu, P., Ke, Q., and Liu, J. Diffusion-based image translation with label guidance for domain adaptive semantic segmentation. In *Proceedings of the IEEE/CVF International Conference on Computer Vision*, pp. 808–820, 2023.

Peng, X., Usman, B., Kaushik, N., Hoffman, J., Wang, D., and Saenko, K. Visda: The visual domain adaptation challenge. *arXiv preprint arXiv:1710.06924*, 2017.

Rangwani, H., Aithal, S. K., Mishra, M., Jain, A., and Radhakrishnan, V. B. A closer look at smoothness in domain adversarial training. In *International conference on machine learning*, pp. 18378–18399. PMLR, 2022.

Richter, S. R., Vineet, V., Roth, S., and Koltun, V. Playing for data: Ground truth from computer games. In *Computer Vision–ECCV 2016: 14th European Conference, Amsterdam, The Netherlands, October 11-14, 2016, Proceedings, Part II 14*, pp. 102–118. Springer, 2016.

Ros, G., Sellart, L., Materzynska, J., Vazquez, D., and Lopez, A. M. The synthia dataset: A large collection of synthetic images for semantic segmentation of urban scenes. In *Proceedings of the IEEE conference on computer vision and pattern recognition*, pp. 3234–3243, 2016.

Shen, F., Gurram, A., Liu, Z., Wang, H., and Knoll, A. Diga: Distil to generalize and then adapt for domain adaptive semantic segmentation. In *Proceedings of the IEEE/CVF Conference on Computer Vision and Pattern Recognition*, pp. 15866–15877, 2023.

Sun, R., Mai, H., Li, W., and Zhang, T. Towards unbiased learning in semi-supervised semantic segmentation. In *The Thirteenth International Conference on Learning Representations*, 2025.

Toldo, M., Michieli, U., Agresti, G., and Zanuttigh, P. Unsupervised domain adaptation for mobile semantic segmentation based on cycle consistency and feature alignment. *Image and Vision Computing*, 95:103889, 2020.

Tranheden, W., Olsson, V., Pinto, J., and Svensson, L. Dacs: Domain adaptation via cross-domain mixed sampling. In *Proceedings of the IEEE/CVF Winter Conference on Applications of Computer Vision*, pp. 1379–1389, 2021.

Truong, T.-D., Le, N., Raj, B., Cothren, J., and Luu, K. Fredom: Fairness domain adaptation approach to semantic scene understanding. In *IEEE/CVF Computer Vision and Pattern Recognition (CVPR)*, 2023.

Tsai, Y.-H., Hung, W.-C., Schulter, S., Sohn, K., Yang, M.-H., and Chandraker, M. Learning to adapt structured output space for semantic segmentation. In *Proceedings of the IEEE conference on computer vision and pattern recognition*, pp. 7472–7481, 2018.

Van Horn, G. and Perona, P. The devil is in the tails: Fine-grained classification in the wild. *arXiv preprint arXiv:1709.01450*, 2017.

Wang, J., Zhang, W., Zang, Y., Cao, Y., Pang, J., Gong, T., Chen, K., Liu, Z., Loy, C. C., and Lin, D. Seesaw loss for long-tailed instance segmentation. In *Proceedings of the IEEE/CVF conference on computer vision and pattern recognition*, pp. 9695–9704, 2021a.

Wang, K., Kim, D., Feris, R., and Betke, M. Cdac: Cross-domain attention consistency in transformer for domain adaptive semantic segmentation. In *Proceedings of the IEEE/CVF International Conference on Computer Vision*, pp. 11519–11529, 2023.

Wang, Y., Peng, J., and Zhang, Z. Uncertainty-aware pseudo label refinery for domain adaptive semantic segmentation. In *Proceedings of the IEEE/CVF international conference on computer vision*, pp. 9092–9101, 2021b.

Wang, Y., Sun, R., Luo, N., Pan, Y., and Zhang, T. Image-to-image matching via foundation models: A new perspective for open-vocabulary semantic segmentation. In *Proceedings of the IEEE/CVF Conference on Computer Vision and Pattern Recognition*, pp. 3952–3963, 2024.

Wangkai, L., Zhaoyang, L., Rui, S., Huayu, M., Naisong, L., Wang, Y., Yuwen, P., Guoxin, X., Huakai, L., Zhiwei, X., et al. Maunet: Modality-aware anti-ambiguity u-net for multi-modality cell segmentation. In *Competitions in Neural Information Processing Systems*, pp. 1–12. PMLR, 2023.

Xie, E., Wang, W., Yu, Z., Anandkumar, A., Alvarez, J. M., and Luo, P. Segformer: Simple and efficient design for semantic segmentation with transformers. *Advances in Neural Information Processing Systems*, 34:12077–12090, 2021.

Yang, L., Hoyer, L., Weber, M., Fischer, T., Dai, D., Leal-Taixé, L., Pollefeys, M., Cremers, D., and Van Gool, L. Micdrop: masking image and depth features via complementary dropout for domain-adaptive semantic segmentation. In *European Conference on Computer Vision*, pp. 329–346. Springer, 2025.

Yu, F., Chen, H., Wang, X., Xian, W., Chen, Y., Liu, F., Madhavan, V., and Darrell, T. Bdd100k: A diverse driving dataset for heterogeneous multitask learning. In *Proceedings of the IEEE/CVF conference on computer vision and pattern recognition*, pp. 2636–2645, 2020.

Zhang, P., Zhang, B., Zhang, T., Chen, D., Wang, Y., and Wen, F. Prototypical pseudo label denoising and target structure learning for domain adaptive semantic segmentation. In *Proceedings of the IEEE/CVF conference on computer vision and pattern recognition*, pp. 12414–12424, 2021.

Zhang, Q., Zhang, J., Liu, W., and Tao, D. Category anchor-guided unsupervised domain adaptation for semantic segmentation. *Advances in neural information processing systems*, 32, 2019.

Zhao, D., Wang, S., Zang, Q., Quan, D., Ye, X., Yang, R., and Jiao, L. Learning pseudo-relations for cross-domain semantic segmentation. In *Proceedings of the IEEE/CVF International Conference on Computer Vision*, pp. 19191–19203, 2023.

Zou, Y., Yu, Z., Liu, X., Kumar, B., and Wang, J. Confidence regularized self-training. In *Proceedings of the IEEE/CVF international conference on computer vision*, pp. 5982–5991, 2019.

## A. Derivation of Eq.7

Considering probabilities estimate that $p(y_{ij} = c|x_{ij}) \propto \exp(f_\theta(x_{ij})[c])$, the discriminant probability for pixel $x_{ij}$ can be presented as:

$$p(y_{ij} = c|x_{ij}) = \frac{\exp(f_\theta(x_{ij})[c]}{\sum_{c'} \exp(f_\theta(x_{ij})[c'])}. \tag{11}$$

Given offset for logits as $\Delta_{cl}(z) = z' - z$ and the label c for pixel $x_{ij}$, we can obtain revised class-conditional discriminant probability for each pixel $x_{ij}$ by:

$$\widetilde{p}(y_{ij} = c|x_{ij}, c) = \frac{\exp(f_\theta(x_{ij})[c] + \Delta_{cc}(f_\theta(x_{ij})[c]))}{\sum_{c'} \exp(f_\theta(x_{ij})[c'] + \Delta_{cc'}(f_\theta(x_{ij})[c']))}. \tag{12}$$

Then, following Bayes rule, the revised posterior is derived as:

$$\widetilde{p}(y_{ij} = c|x_{ij}) = \frac{p(c)\widetilde{p}(y_{ij} = c|x_{ij}, c)}{\sum_{c'} p(c')\widetilde{p}(y_{ij} = c|x_{ij}, c')}. \tag{13}$$

So we can obtain revised prediciton results for each pixel $x_{ij}$ by:

$$\widetilde{y}_{ij} = \text{argmax}_{c \in [C]} p(c)\widetilde{p}(y_{ij} = c|x_{ij}, c). \tag{14}$$

Since the class probabilities $p(c)$ are typically set as a uniform prior, i.e., $p(c) = \frac{1}{c}$, Eq.14 can be rewritten as:

$$\widetilde{y}_{ij} = \text{argmax}_{c \in [C]} p(y_{ij} = c|x_{ij}, c)$$
$$= \text{argmax}_{c \in [C]} \frac{\exp(f_\theta(x_{ij})[c] + \Delta_{cc}(f_\theta(x_{ij})[c]))}{\exp(f_\theta(x_{ij})[c] + \Delta_{cc}(f_\theta(x_{ij})[c])) + \sum_{c' \neq c} \exp(f_\theta(x_{ij})[c'] + \Delta_{cc'}(f_\theta(x_{ij})[c']))}. \tag{15}$$

For Eq.6, we add a scaling factor $\tau$ to be adjusted for specific evaluation metrics.

## B. Discussion about Eq.9

For a deeper understanding of the loss function in Eq.9, we can rewrite it as:

$$\widetilde{\mathcal{L}}^s = \frac{1}{N_S} \sum_{i=1}^{N_S} \sum_{j=1}^{H \times W} \log \left( 1 + \sum_{c \neq y_{ij}^S} \left( \frac{\exp\left(\Delta_{y_{ij}^S, y_{ij}^S}^S (f_\theta(x_{ij}^S)[y_{ij}^S])\right)}{\exp\left(\Delta_{y_{ij}^S, c}^S (f_\theta(x_{ij}^S)[c])\right)} \right)^\tau \exp\left(f_\theta(x_{ij}^S)[c] - f_\theta(x_{ij}^S)[y_{ij}^S]\right) \right), \tag{16}$$

which can be interpreted as a standard cross-entropy loss with an adaptive margin. Specifically, if the class $y_{ij}^S$ is an over-predicted class, it is reasonable to assume that for most other classes $c$, $\Delta_{y_{ij}^S, y_{ij}^S}^S (f_\theta(x_{ij}^S)[y_{ij}^S]) < \Delta_{y_{ij}^S, c}^S (f_\theta(x_{ij}^S)[c])$. This implies that:

$$\left( \frac{\exp\left(\Delta_{y_{ij}^S, y_{ij}^S}^S (f_\theta(x_{ij}^S)[y_{ij}^S])\right)}{\exp\left(\Delta_{y_{ij}^S, c}^S (f_\theta(x_{ij}^S)[c])\right)} \right)^\tau < 1 \tag{17}$$

In the loss function Eq.16, this ratio is used to scale the term $\exp\left(f_\theta(x_{ij}^S)[c] - f_\theta(x_{ij}^S)[y_{ij}^S]\right)$. When $y_{ij}^S$ is an over-predicted class, the ratio is less than 1, which reduces the weight of the term $\exp\left(f_\theta(x_{ij}^S)[c] - f_\theta(x_{ij}^S)[y_{ij}^S]\right)$. Consequently, for over-predicted classes, the loss function imposes a smaller penalty for misclassification. Conversely, if $y_{ij}^S$ is an under-predicted class, it can be assumed that for most other classes $c$, $\Delta_{y_{ij}^S, y_{ij}^S}^S (f_\theta(x_{ij}^S)[y_{ij}^S]) > \Delta_{y_{ij}^S, c}^S (f_\theta(x_{ij}^S)[c])$. This leads to a ratio greater than 1, which increases the weight of the term $\exp\left(f_\theta(x_{ij}^S)[c] - f_\theta(x_{ij}^S)[y_{ij}^S]\right)$, thereby imposing a larger penalty for misclassification of under-predicted classes. In summary, by introducing the margin-based ratio term, the loss function Eq.16 adaptively adjusts the strength of the penalty based on the difficulty of the classes. This approach helps to mitigate the class bias problem and enhances the model's performance on under-predicted classes, leading to a more balanced and accurate classification. Moreover, since the logit offset term $\Delta$ is updated online during the training process, it aligns well with the self-training paradigm in UDA.

## C. Discussion about Evaluation Metrics

In this section, we investigate the impact of class imbalance on the evaluation metrics mIoU and mAcc in the context of semantic segmentation. We consider a multi-class problem with $C$ classes, where the number of samples in the $i$-th class is denoted as $N_i$. Let $P_{ij}$ represent the probability of classifying a sample from the $i$-th class as the $j$-th class in the confusion matrix. For the purpose of this analysis, we assume that the probabilities $P_{ii}$ and $P_{kk}$ are balanced across all classes, i.e., $P_{ii} = P_{kk} = p, \forall i, k \in 1, 2, ..., C$, and focus solely on the effect of class imbalance in terms of sample numbers. Under this assumption, the calculation formula for mAcc can be written as:

$$mAcc = \frac{1}{C} \sum_{i=1}^{C} \frac{N_i \cdot P_{ii}}{\sum_{j=1}^{C} N_i \cdot P_{ij}} = \frac{1}{C} \sum_{i=1}^{C} \frac{p}{\sum_{j=1}^{C} P_{ij}} \tag{18}$$

As evident from the equation above, the sample numbers $N_i$ cancel out in the calculation of mAcc, making it independent of the class imbalance in terms of sample numbers. Therefore, mAcc remains unaffected by class imbalance under the given assumptions. On the other hand, the calculation formula for mIoU is given by:

$$mIoU = \frac{1}{C} \sum_{i=1}^{C} \frac{N_i \cdot P_{ii}}{\sum_{j=1}^{C} N_i \cdot P_{ij} + \sum_{j=1}^{C} N_j \cdot P_{ji} - N_i \cdot P_{ii}} \tag{19}$$

In contrast to mAcc, the sample numbers $N_i$ do not cancel out in the calculation of mIoU. Consequently, when the classes are imbalanced, i.e., the sample numbers $N_i$ vary significantly across classes, the IoU of classes with larger sample numbers will dominate the overall mIoU result. To illustrate the impact of class imbalance on mIoU, let us consider a case where class $k$ is a head class with a significantly larger sample number $N_k$ compared to a tail class $i$ with sample number $N_i$. The performance of class $i$ will be greatly affected by class $k$ through the term $N_k \cdot P_{ki}$ in the denominator of mIoU. For class $i$ to have a fair contribution to mIoU, the probability $P_{ki}$ needs to be very small. This implies that a balanced classifier may not be optimal for maximizing mIoU under class imbalance. Therefore, our proposed method implements a scaling factor $\tau$ to modulate the contributions of introduced logits offset. In contrast, mAcc is inherently unaffected by class imbalance, as the sample numbers $N_i$ cancel out in its calculation formula. This means that a balanced classifier is indeed optimal for maximizing mAcc, and our method, which aims to balance the contributions of different classes, aligns well with this objective and can achieve consistent improvements.

## D. Implementation Details of Online Logits Distribution Estimation

In this section, we provide the pseudo label to explain implementation details of online logits distribution estimation, as shown in Alg.1. For computational efficiency, in each iteration, we sample $N_{sample}$ logits for each element in $M_{cl}^S$ and $M_{cl}^T$, where $N_{sample}$ is the minimum sample number of classes in the minibatch (shoule be greater than or equal to $N_{min}$, where we set it as 100). This way, we obtain $C \times C \times N_{sample}$ logits, and then update the $C \times C$ GMMs simultaneously in a parallel manner. Since not all classes may be updated in each iteration, we maintain a variable $n$ for each GMM that is not updated, to record the number of iterations since its last update. This $n$ is used to adjust the momentum factor using $\tilde{\tau}$ in the EMA update, in order to match the update speed of the network.

In the algorithm, the cumulative distribution functions (CDFs) $F_{cl}^S$, $F_{cl}^T$, $F_p$, and $F_n$ are computed using the estimated Gaussian mixture models (GMMs). These CDFs describe the cumulative probability distribution of the corresponding GMMs. For the source and target domain GMMs $P_{cl}^S$ and $P_{cl}^T$, their CDFs can be represented as: $F_{cl}^S(z) = \sum_{k=1}^{K} \pi_{cl,k}^S \cdot \Phi(\frac{z - \mu_{cl,k}^S}{\sigma_{cl,k}^s})$ $F_{cl}^T(z) = \sum_{k=1}^{K} \pi_{cl,k}^T \cdot \Phi(\frac{z - \mu_{cl,k}^T}{\sigma_{cl,k}^t})$, where $\Phi(\cdot)$ is the CDF of the standard normal distribution, and $\pi_{cl,k}^S$, $\mu_{cl,k}^S$, and $\sigma_{cl,k}^S$ are the weight, mean, and standard deviation of the $k$-th component of the source domain GMM $P_{cl}^S$, respectively. $\pi_{cl,k}^T$, $\mu_{cl,k}^T$, and $\sigma_{cl,k}^T$ are the corresponding parameters for the target domain GMM $P_{cl}^T$. Similarly, the CDFs for the anchor GMMs $P_p$ and $P_n$ are: $F_p(z) = \sum_{k=1}^{K} \pi_{p,k} \cdot \Phi(\frac{z - \mu_{p,k}}{\sigma_{p,k}})$ $F_n(z) = \sum_{k=1}^{K} \pi_{n,k} \cdot \Phi(\frac{z - \mu_{n,k}}{\sigma_{n,k}})$. The inverse function of a CDF, denoted as $F^{-1}(\cdot)$, represents the value of the variable corresponding to a given cumulative probability. For a given logit value $z$, by computing $F_p^{-1}(F_{cl}^S(z))$ and $F_n^{-1}(F_{cl}^S(z))$, we obtain the corresponding logit values of the positive anchor distribution $P_p$ and the negative anchor distribution $P_n$ at the cumulative probability $F_{cl}^S(z)$. Then, the difference between these values and the original logit value $z$ is used as the logits offset $\Delta_{cl}^S(z)$ for the source domain. Similarly, by computing

---

**Algorithm 1** Online Logits Adjustment for UDA

---

**Require:** Source domain $D_S = (x_i^S, y_i^S)_{i=1}^{N_S}$, target domain $D_T = (x_i^T)_{i=1}^{N_T}$, number of classes $C$, number of Gaussian
   components $K$, momentum factor $\tilde{\tau}$, scaling factor $\tau$, minimum number of elements $N_{min}$, number of EM Loop $\mathcal{T}$.
**Ensure:** Model parameters $\theta$.
 1: Initialize model parameters $\theta$, source GMMs $P_{cl}^S$, target GMMs $P_{cl}^T$, anchor GMMs $P_p$ and $P_n$ for all $c, l \in [C]$.
 2: **while** not converged **do**
 3:     Sample a mini-batch of source data $(x_i^S, y_i^S)_{i=1}^{B_S}$ and target data $(x_i^T)_{i=1}^{B_T}$.
 4:     Compute logits $f_\theta(x_{ij}^S)$ and $f_\theta(x_{ij}^T)$ for source and target samples.
 5:     Compute pseudo-labels for target samples: $\hat{y}_{ij}^T = \operatorname{argmax}_{c \in [C]} f_\theta(x_{ij}^T[c])$.
 6:     Compute matrices $M_{cl}^S$ and $M_{cl}^T$ based on the source labels and target pseudo-labels:
 7:     $M_{cl}^S = \{f_\theta(x_{ij}^S)[l] \mid y_{ij}^S = c\}$
 8:     $M_{cl}^T = \{f_\theta(x_{ij}^T)[l] \mid \hat{y}_{ij}^T = c\}$
 9:     Update source GMMs $P_{cl}^S$, target GMMs $P_{cl}^T$, anchor GMMs $P_p$ and $P_n$ using the momentum-based EM algorithm:
10:     **for** $c, l \in [C]$ **do**
11:         **if** $|M_{cl}^S| > N_{min}$ **then**
12:             Initialize $\phi_{cl}^{S,(0)} \leftarrow \hat{\phi}_{cl}^S$ from the latest iteration.
13:             **for** $t = 1, \ldots, \mathcal{T}$ **do**
14:                 Update $\phi_{cl}^{S,(t)}$ using the EM algorithm with current logits.
15:             **end for**
16:             $\hat{\phi}_{cl}^S \leftarrow (1 - \tilde{\tau}^n)\phi_{cl}^{S,(\mathcal{T})} + \tilde{\tau}^n \hat{\phi}_{cl}^S$, where $n$ is the number of iterations since the last update.
17:         **end if**
18:         **if** $|M_{cl}^S| > N_{min}$ **then**
19:             Initialize $\phi_{cl}^{T,(0)} \leftarrow \hat{\phi}_{cl}^T$ from the latest iteration.
20:             **for** $t = 1, \ldots, \mathcal{T}$ **do**
21:                 Update $\phi_{cl}^{T,(t)}$ using the EM algorithm with current logits.
22:             **end for**
23:             $\hat{\phi}_{cl}^T \leftarrow (1 - \tilde{\tau}^n)\phi_{cl}^{T,(\mathcal{T})} + \tilde{\tau}^n \hat{\phi}_{cl}^T$, where $n$ is the number of iterations since the last update.
24:         **end if**
25:     **end for**
26:     Update anchor GMMs $P_p$ and $P_n$ using the global positive and negative logits from the source domain:
27:     Initialize $\phi_p^{(0)} \leftarrow \hat{\phi}_p$, $\phi_n^{(0)} \leftarrow \hat{\phi}_n$ from the latest iteration.
28:     **for** $t = 1, \ldots, \mathcal{T}$ **do**
29:         Update $\phi_p^{(t)}$, $\phi_n^{(t)}$ using the EM algorithm with current global logits.
30:     **end for**
31:     $\hat{\phi}_p \leftarrow (1 - \tilde{\tau})\phi_p^{(\mathcal{T})} + \tilde{\tau}\hat{\phi}_p$
32:     $\hat{\phi}_n \leftarrow (1 - \tilde{\tau})\phi_n^{(\mathcal{T})} + \tilde{\tau}\hat{\phi}_n$
33:     Compute cumulative distributions $F_{cl}^S$, $F_{cl}^T$, $F_p$, $F_n$ using the estimated GMMs.

34:     Compute logits offsets for source domain: $\Delta_{cl}^S(z) = \begin{cases} F_p^{-1}(F{cl}^S(z)) - z, & \text{if } c = l \\ F_n^{-1}(F{cl}^S(z)) - z, & \text{if } c \neq l \end{cases}$

35:     Compute logits offsets for target domain: $\Delta_{cl}^T(z) = \begin{cases} F_p^{-1}(F_{cl}^T(z)) - z, & \text{if } c = l \\ F_n^{-1}(F_{cl}^T(z)) - z, & \text{if } c \neq l \end{cases}$

36:     Compute the adjusted losses $\widetilde{\mathcal{L}}^s$ and $\widetilde{\mathcal{L}}^u$ using Eq. (8) with $\Delta_{cl}^S$ and $\Delta_{cl}^T$.
37:     Update model parameters $\theta$ by minimizing $\widetilde{\mathcal{L}}^s + \widetilde{\mathcal{L}}^u$ using an optimizer (e.g., SGD or Adam).
38: **end while**
39: **return** Model parameters $\theta$.

---

$F_p^{-1}(F_{cl}^T(z))$ and $F_n^{-1}(F_{cl}^T(z))$, we obtain the logits offset $\Delta_{cl}^T(z)$ for the target domain. To efficiently compute the CDF and its inverse function, we use the Abramowitz-Stegun formula to approximate the CDF in the form of a polynomial and employ interpolation methods to estimate the inverse function.

Table 5: Parameter study of $K$.

| $K$ | mIou (%) |
|---|---|
| 1 | 70.2 |
| 3 | 70.5 |
| 5 | **70.7** |
| 10 | 70.6 |

Table 6: Parameter study of $\tilde{\tau}$.

| $\tilde{\tau}$ | mIou (%) |
|---|---|
| 0 | 68.6 |
| 0.9 | 70.0 |
| 0.99 | **70.7** |
| 0.999 | 70.3 |

Table 7: Parameter study of $\mathcal{T}$.

| $\mathcal{T}$ | mIou (%) |
|---|---|
| 1 | 70.4 |
| 3 | **70.7** |
| 5 | **70.7** |
| 10 | 70.5 |

Table 8: Parameter study of $\lambda$.

| $\lambda$ | mIou (%) |
|---|---|
| 0.05 | 70.3 |
| 0.2 | **70.7** |
| 0.5 | 70.4 |
| 1 | 69.9 |

## E. Influence of Parameters Setting

In this section, we further study the influence of parameters setting introduced in BLDA, i.e., number of Gaussian components $K$, momentum factor $\tilde{\tau}$, number of EM Loop $\mathcal{T}$ for GMM estimation and $\lambda$ for cumulative density estimation loss. All experiments are conducted with DAFormer (Hoyer et al., 2022a) on GTA→CS.

**Number of Gaussian Components $K$.** As shown in Tab.5, we find that BLDA can work well even when $K = 1$, since the logit distribution is naturally close to Gaussian. The model achieves the best performance when $K = 5$. A larger $K$ allows for more flexibility in modeling the logits distributions but may also introduce noise. We choose $K = 5$ as a balance between model capacity and robustness.

**Momentum Factor $\tilde{\tau}$.** The momentum factor $\tilde{\tau}$ controls the speed of updating the GMM parameters. When $\tilde{\tau} = 0$, performance becomes erratic because the logits from the current iteration alone are not sufficient to model the distribution of all logits. A larger $\tilde{\tau}$ leads to slower updates, retaining the previously estimated distribution and making the estimation more stable but less adaptive. As presented in Tab.6, setting $\tilde{\tau}$ to 0.99 yields the best performance, suggesting that a relatively stable estimation of the logits distributions is beneficial for the adaptation process.

**Number of EM Loop $\mathcal{T}$.** The number of EM loops $\mathcal{T}$ determines the number of iterations used to update the GMM parameters in each training step. Tab.7 shows that the model is not sensitive to the choice of $\mathcal{T}$, since the convergence rate of GMM is faster than the rate of network update, and it can be estimated well even when $\mathcal{T} = 1$. We choose $\mathcal{T} = 3$ for stable performance while considering computational efficiency.

**Cumulative Density Estimation Loss Weight $\lambda$.** The weight $\lambda$ balances the cumulative density estimation loss with the segmentation loss. A higher $\lambda$ enforces stronger domain alignment through the cumulative density functions. As shown in Tab.8, $\lambda = 0.2$ provides the best performance gain. An overly large $\lambda$ may distract the model from learning the primary segmentation task, leading to performance degradation.

## F. Additional Experiment Results

In Table 9, we report additional Acc metric for Table 2. The improvement with mAcc in this benchmark is also more significant, which aligns with our expectations.

Table 9: UDA segmentation performance on SYN.→CS. using the mAcc (%) evaluation metric, where the improvement is marked as **bold**.

| Method | Arch. | Road | Sidewalk | Building | Wall | Fence | Pole | Light | Sign | Veg | Terrain | Sky | Person | Rider | Car | Truck | Bus | Train | Motor | Bike | mAcc (↑) | std (↓) |
|---|---|---|---|---|---|---|---|---|---|---|---|---|---|---|---|---|---|---|---|---|---|---|
| DAFormer(Hoyer et al., 2022a) | T | 89.8 | 90.2 | 96.2 | 33.8 | 8.3 | 51.9 | 63.1 | 57.8 | 95.1 | - | 98.4 | 86.3 | 63.6 | 96.7 | - | 83.4 | - | 55.4 | 61.4 | 70.7 | 25.1 |
| +BLDA | T | 87.3 | **95.6** | 94.9 | **38.8** | **14.1** | **57.5** | **70.1** | 63.3 | 97.8 | - | **98.9** | **90.0** | **68.0** | 97.7 | - | **95.7** | - | 63.8 | 67.5 | 75.1 | 23.6 |
| HRDA (Hoyer et al., 2022b) | T | 90.3 | 85.2 | 96.0 | 69.4 | 8.0 | 70.6 | 81.2 | 69.6 | 94.7 | - | 99.1 | 88.0 | 68.9 | 97.4 | - | 93.8 | - | 71.3 | 74.0 | 78.6 | 21.2 |
| +BLDA | T | 89.4 | **94.5** | **95.4** | 75.5 | 24.5 | 78.2 | 85.8 | 74.7 | 97.6 | - | 98.8 | **88.9** | 71.6 | 97.4 | - | **96.9** | - | **72.6** | **76.2** | 82.4 | 17.9 |
| MIC (Hoyer et al., 2023) | T | 89.9 | 87.4 | 96.2 | 71.3 | 8.4 | 66.7 | 81.5 | 69.7 | 95.6 | - | 98.5 | 89.1 | 73.0 | 97.3 | - | 93.5 | - | 78.2 | 71.5 | 79.2 | 21.2 |
| +BLDA | T | 89.4 | **97.2** | 96.0 | **73.4** | **17.9** | **72.1** | **87.6** | **75.8** | 97.4 | - | 98.3 | **91.5** | 72.5 | **97.7** | - | **96.2** | - | **79.3** | 75.6 | **82.4** | **19.4** |

## G. Extended Experiment on Image Classification

To demonstrate the generality of BLDA, we implement BLDA based on MIC (Hoyer et al., 2023) with ResNet-101 on the VisDA-2017 (Peng et al., 2017) UDA classification benchmark in Tab.10, and our method still achieves improvements. In the classification task, the dataset does not have severe class distribution differences like segmentation. However, as we

point out in Fig.1, the transfer difficulty differences between domains still lead to severe class bias in this task, and our method can effectively alleviate this and achieve more balanced predictions.

Table 10: Image classification accuracy in % on VisDA-2017 for UDA, where the improvement is marked as **bold**.

| Method | Plane | Bcycl | Bus | Car | Horse | Knife | Mcyle | Persn | Plant | Sktb | Train | Truck | Mean (↑) | Std (↓) |
|---|---|---|---|---|---|---|---|---|---|---|---|---|---|---|
| MCC (Jin et al., 2020) | 88.1 | 80.3 | 80.5 | 71.5 | 90.1 | 93.2 | 85.0 | 71.6 | 89.4 | 73.8 | 85.0 | 36.9 | 78.8 | 14.4 |
| SDAT (Rangwani et al., 2022) | 95.8 | 85.5 | 76.9 | 69.0 | 93.5 | 97.4 | 88.5 | 78.2 | 93.1 | 91.6 | 86.3 | 55.3 | 84.3 | 11.9 |
| MIC (Hoyer et al., 2023) | 96.7 | 88.5 | 84.2 | 74.3 | 96.0 | 96.3 | 90.2 | 81.2 | 94.3 | 95.4 | 88.9 | 56.6 | 86.9 | 11.3 |
| **+BLDA** | 96.2 | **90.3** | 82.8 | **81.2** | 95.7 | **96.7** | **93.4** | **86.5** | **95.7** | 94.3 | **91.0** | **65.7** | **89.1** | **8.7** |

# H. Extended Experiment on Video Semantic Segmentation

We further validate our method on the Domain Adaptive Video Semantic Segmentation setting. Table 11 shows results on VIPER (Richter et al., 2016) → Cityscapes-Seq, and Table 12 shows results on VIPER → BDD (Yu et al., 2020), where we divided the BDD100k subset following (Kareer et al., 2024). All experiments are based on the Deeplabv2 (Chen et al., 2017a) segmentation network.

Table 11: Domain adaptive video semantic segmentation performance on VIPER → Cityscapes-Seq.

| Method | Road | Sidewalk | Building | Fence | Light | Sign | Veg | Terrain | Sky | Person | Car | Truck | Bus | Motor | Bike | mIoU (↑) | std (↓) |
|---|---|---|---|---|---|---|---|---|---|---|---|---|---|---|---|---|---|
| PAT (Mai et al., 2024a) | 89.3 | 45.5 | 83.5 | 27.5 | 35.7 | 36.1 | 86.6 | 34.8 | 85.9 | 63.0 | 86.7 | 37.9 | 46.9 | 29.3 | 29.9 | 54.5 | 24.0 |
| **+BLDA** | 88.9 | **61.0** | **83.8** | **30.1** | **45.6** | **38.5** | 84.3 | **35.1** | 84.7 | 61.4 | 84.2 | **40.5** | **48.2** | 29.7 | **33.6** | **56.6** | **22.1** |
| MIC (Hoyer et al., 2023) | 95.5 | 71.5 | 91.2 | 21.1 | 66.2 | 73.1 | 89.8 | 47.3 | 92.3 | 79.6 | 89.7 | 48.6 | 54.9 | 38.5 | 64.6 | 68.3 | 21.7 |
| **+BLDA** | 95.1 | **73.1** | 90.7 | **29.9** | **69.6** | **74.2** | 89.6 | **48.9** | 90.5 | 77.9 | **89.8** | **50.1** | **56.7** | **40.7** | **69.1** | **69.7** | **19.7** |

Table 12: Domain adaptive video semantic segmentation performance on VIPER → BDD.

| Method | Road | Sidewalk | Building | Fence | Light | Sign | Veg | Terrain | Sky | Person | Car | Truck | Bus | Motor | Bike | mIoU (↑) | std (↓) |
|---|---|---|---|---|---|---|---|---|---|---|---|---|---|---|---|---|---|
| HRDA (Hoyer et al., 2022b) | 84.2 | 35.5 | 80.7 | 13.1 | 24.0 | 28.0 | 75.1 | 25.0 | 84.1 | 60.4 | 84.4 | 37.8 | 44.7 | 30.6 | 29.7 | 49.2 | 25.2 |
| **+BLDA** | **84.4** | **40.1** | 79.5 | **18.7** | **27.3** | **29.1** | 74.7 | **25.5** | 82.6 | 62.8 | 81.7 | **43.9** | **46.7** | **35.1** | **29.9** | **50.8** | **23.4** |

# I. BLDA Efficiency Analysis

In our experiments, our method is highly efficient in terms of both training time and computational cost. We have provided a detailed explanation of our implementation in Appendix D. The additional cost introduced by our proposed components can be attributed to the following factors, for which we have implemented efficient solutions:

- **GMM implementation**: Instead of using off-the-shelf libraries to update the parameters sequentially, we store the parameters of $C \times C \times K$ Gaussian components as tensors in PyTorch and perform parallel updates.

- **CDF computation**: We use the Abramowitz-Stegun formula to approximate the CDF in the form of a polynomial.

- **Inverse CDF computation**: We use interpolation algorithms to approximate the inverse CDF within the estimated range of values.

All the above operations can be efficiently performed using simple matrix operations on tensors. Moreover, the storage of Gaussian component parameters and the additional regression head (a $1 \times 1$ conv) introduced by our method are lightweight. We also report the training time and computational cost of our method when applied to different methods, as shown in the Table 13. Our method is computationally efficient, with an average increase of 0.3s per iteration for each method (all methods are trained with a batch size of 2 and an image crop size of $512 \times 512$). The total additional time depends on the total number of iterations for each method. For DAFormer and DACS, we observe an increase of approximately 3GB in GPU memory usage, while for CDAC, no additional memory consumption is observed (possibly because our module reuses the cached memory already allocated by this method).

Table 13: Computational resource requirements comparison

| Methods | GPU Memory (MB) | Time per iter (s) | Total Time (h) |
|---|---|---|---|
| DAFormer (Hoyer et al., 2022a) | 9,807 | 1.32 | 14.5 (40K iters) |
| **+BLDA** | 12,655 | 1.59 | 17.7 (40K iters) |
| DACS (Tranheden et al., 2021) | 11,078 | 0.52 | 35.5 (250K iters) |
| **+BLDA** | 14,354 | 0.81 | 56.1 (250K iters) |
| CDAC (Wang et al., 2023) | 35,443 | 1.66 | 18.7 (40K iters) |
| **+BLDA** | 35,443 | 1.95 | 22.3 (40K iters) |

## J. Further Discussion With Anchor Distribution

In our experiments, we use anchor distribution estimated from source domain to balance both source and target domains. Normally, there exists a discrepancy between the anchor distribution and the true distribution of the target domain. However, the impact of this distributional difference on the final performance is relatively small because the relative difference between the positive and negative distributions plays a more crucial role in the anchor distribution, while the absolute difference in their overall distributions (e.g., simultaneously increasing/decreasing pos/neg) does not significantly affect the performance. In Fig.1(c), we observe that the biases of positive and negative logits are coupled, meaning they tend to be either both larger or smaller. The differences in bias distributions are more reflected in the absolute differences of the overall distribution, while the relative differences between pos/neg do not vary significantly. Intuitively, if there is a distributional bias between the target domain and the source domain, this difference would also be more evident in the overall absolute difference.

For empirical results, we conduct an ablation study on the selection criteria for these anchor distributions, considering both post-hoc and online logits adjustment (OLA) implementations to observe the results in Table 14. In our setting, the anchor distribution is obtained from the global distribution statistics of the source domain. For the post-hoc implementation, we consider the global distribution of the target domain and the pos/neg logits distributions of two specific classes (building and fence, which are the two most biased classes in Fig. 1(d)) as anchors. We find that the impact of different anchor distribution choices on the final performance is relatively small.

For the OLA implementation, we find that using the global logits distribution is important because our online distribution estimation approach leads to smaller estimation errors when using global pos/neg statistics, promoting stable training during the self-training process.

Table 14: Ablation on different selection criteria for anchor distributions on GTA.→CS. using the mIoU (%) evaluation metric. OLA denotes online logits adjustment.

| Anchor | Strategy | Road | Sidewalk | Building | Wall | Fence | Pole | Light | Sign | Veg | Terrain | Sky | Person | Rider | Car | Truck | Bus | Train | Motor | Bike | mIoU (↑) | std (↓) |
|---|---|---|---|---|---|---|---|---|---|---|---|---|---|---|---|---|---|---|---|---|---|---|
| DAFormer (Hoyer et al., 2022a) | - | 95.7 | 70.2 | 89.4 | 53.5 | 48.1 | 49.6 | 55.8 | 59.4 | 89.9 | 47.9 | 92.5 | 72.2 | 44.7 | 92.3 | 74.5 | 78.2 | 65.1 | 55.9 | 61.8 | 68.3 | 16.8 |
| target (global) | post-hoc | 95.7 | 77.0 | 87.8 | 61.0 | 53.7 | 54.3 | 56.1 | 61.2 | 87.2 | 50.2 | 90.4 | 74.4 | 43.4 | 90.9 | 73.3 | 82.5 | 56.6 | 54.8 | 68.3 | 69.4 | 15.8 |
| source (building) | post-hoc | 95.8 | 77.1 | 90.0 | 59.8 | 54.5 | 51.6 | 56.6 | 59.8 | 86.2 | 48.2 | 90.6 | 75.4 | 43.7 | 90.4 | 72.0 | 79.5 | 58.9 | 53.7 | 69.2 | 69.1 | 16.0 |
| source (fence) | post-hoc | 95.6 | 77.5 | 88.0 | 58.4 | 55.0 | 51.2 | 56.9 | 57.7 | 88.7 | 48.3 | 90.5 | 75.2 | 43.8 | 90.1 | 71.7 | 80.6 | 61.1 | 53.9 | 65.5 | 68.9 | 16.0 |
| source (global) | post-hoc | 95.7 | 77.1 | 88.6 | 60.5 | 55.3 | 48.5 | 57.3 | 60.9 | 89.5 | 47.2 | 91.0 | 72.9 | 43.7 | 91.3 | 73.7 | 80.8 | 61.1 | 55.3 | 63.8 | 69.2 | 16.2 |
| source (building) | OLA | 95.5 | 77.9 | 88.5 | 60.4 | 59.1 | 53.3 | 57.9 | 60.4 | 88.3 | 49.2 | 90.3 | 76.1 | 43.6 | 90.3 | 76.4 | 85.7 | 56.2 | 57.9 | 68.8 | 70.3 | 15.8 |
| source (fence) | OLA | 95.6 | 81.2 | 88.2 | 57.9 | 57.0 | 51.6 | 54.2 | 60.2 | 87.9 | 50.2 | 90.3 | 76.3 | 44.3 | 90.0 | 75.0 | 82.4 | 57.7 | 56.0 | 70.6 | 69.8 | 16.0 |
| source (global) | OLA | 95.4 | 78.3 | 88.3 | 54.0 | 55.2 | 55.7 | 60.3 | 65.2 | 89.2 | 47.3 | 91.1 | 71.4 | 44.8 | 91.6 | 74.3 | 83.4 | 73.2 | 59.3 | 67.1 | 70.7 | 15.5 |

Furthermore, our anchor distribution serves two purposes during the training process, as discussed in Sec.3.3.4: Using a shared anchor distribution allows the logits distribution of the target domain to gradually align with the source domain. In this implementation, the anchor distribution can both serve as a reference distribution to balance the learning progress between classes within a domain and act as a bridge to connect the two domains at the same time.

To further illustrate this point, please refer to Appendix B. Our logits adjustment can be interpreted as a standard cross-entropy loss with an adaptive margin. We can discuss two cases: 1. If the pos/neg of the anchor simultaneously increase/decrease, this margin remains unchanged, consistent with our conclusion above. 2. If the relative difference between pos/neg in the target domain is greater/smaller than the relative difference between pos/neg in the anchor, then this margin will correspondingly decrease/increase, ultimately leading to the alignment of the distribution with the anchor.Experimentally, we calculate the differences between the distribution of each class in the target domain and the anchor distribution before and after applying our method, using JS divergence as shown in Table 15. This further demonstrates that our online logits adjustment gradually

aligns the target logits distribution with the anchor.

Table 15: The JS divergence between the $P_{cl}$ and corresponding anchor distribution $P_p$ and $P_n$, where the results on $P_n$ is averaged over the $C-1$ negative components.

| Method | Anchor | Road | Sidewalk | Building | Wall | Fence | Pole | Light | Sign | Veg | Terrain | Sky | Person | Rider | Car | Truck | Bus | Train | Motor | Bike | mean (↑) | std (↓) |
|---|---|---|---|---|---|---|---|---|---|---|---|---|---|---|---|---|---|---|---|---|---|---|
| baseline | $P_p$ | 0.412 | 0.289 | 0.381 | 0.286 | 0.293 | 0.137 | 0.325 | 0.142 | 0.429 | 0.273 | 0.596 | 0.130 | 0.217 | 0.300 | 0.225 | 0.164 | 0.190 | 0.145 | 0.180 | 0.270 | 0.119 |
| **+BLDA** | $P_p$ | 0.367 | 0.246 | 0.261 | 0.153 | 0.212 | 0.221 | 0.146 | 0.234 | 0.281 | 0.173 | 0.310 | 0.244 | 0.070 | 0.273 | 0.284 | 0.181 | 0.125 | 0.207 | 0.125 | 0.217 | 0.072 |
| baseline | $P_n$ | 0.517 | 0.327 | 0.315 | 0.394 | 0.305 | 0.280 | 0.422 | 0.288 | 0.412 | 0.404 | 0.433 | 0.278 | 0.396 | 0.279 | 0.387 | 0.382 | 0.401 | 0.430 | 0.394 | 0.371 | 0.064 |
| **+BLDA** | $P_n$ | 0.231 | 0.165 | 0.153 | 0.135 | 0.209 | 0.118 | 0.224 | 0.119 | 0.248 | 0.246 | 0.240 | 0.186 | 0.158 | 0.170 | 0.199 | 0.208 | 0.176 | 0.162 | 0.136 | 0.183 | 0.041 |

## K. Comparison With Other Class-imbalanced Methods

In Fig.2, we show the impact of resampling and reweighting methods on self-training. To further compare with these class-imbalanced methods, we adopt several common approaches, as shown in Table 16.

Table 16: Comparison with other class-imbalanced methods.

| | [1] (Lin et al., 2017) | [2] (Cui et al., 2019) | [3] (Hoyer et al., 2022a) | [4] (Menon et al., 2020) | Ours |
|---|---|---|---|---|---|
| **Task** | Object Detection | Classification | Segmentation | Classification | Segmentation |
| **Motivation** | Reweighting | Reweighting | Resampling | Logit adjustment | Logit adjustment |
| **Implementation** | Weight samples based on confidence, with harder samples receiving higher weights | Define effective number of samples for each class to weight different classes | Sample based on prior knowledge of class distribution, with more sampling for rare classes | Introduce correction terms in logits, determined by prior knowledge of class distribution | 1. Evaluate class bias through logits sets in a form of confusion matrix. 2. Explicitly balance components in the matrix through anchor distributions and cumulative density functions, implemented in an online self-training paradigm. |
| **Note** | | | | | **No prior knowledge of class distribution is introduced**. Class bias is directly modeled based on actual logits distribution. |

Furthermore, we conduct comparative experiments in Table 17. We further discuss more class-imbalanced methods. Please refer to Appendix P for details.

## L. Groups of Over-prediction/Under-prediction

We divide the classes into two groups: over-prediction and under-prediction (refer to Fig.1(d)). Then, for our experiments, we report the performance of the two groups as shown in Table 18. We observe that the main performance gains come from

Table 17: Comparison with other class-imbalanced methods on GTA.→CS. using the mAcc (%) evaluation metric, where baseline is based on DAFormer with uniform class sampling.

| Method | Arch. | Road | Sidewalk | Building | Wall | Fence | Pole | Light | Sign | Veg | Terrain | Sky | Person | Rider | Car | Truck | Bus | Train | Motor | Bike | mAcc (↑) | std (↓) |
|---|---|---|---|---|---|---|---|---|---|---|---|---|---|---|---|---|---|---|---|---|---|---|
| baseline | T | 98.3 | 75.4 | 94.9 | 53.8 | 57.8 | 53.6 | 70.2 | 59.7 | 96.2 | 57.6 | 99.1 | 83.5 | 67.6 | 95.5 | 87.1 | 84.3 | 73.8 | 74.5 | 77.5 | 76.9 | 15.3 |
| [1] (Lin et al., 2017) | T | 91.7 | 63.3 | 94.6 | 66.4 | 50.2 | 54.4 | 63.9 | 55.1 | 95.6 | 66.9 | 98.6 | 84.9 | 58.5 | 92.6 | 84.4 | 81.9 | 70.0 | 68.8 | 68.2 | 74.2 | 15.2 |
| [2] (Cui et al., 2019) | T | 98.8 | 77.7 | 94.1 | 71.4 | 56.8 | 63.7 | 77.2 | 71.2 | 95.3 | 63.5 | 99.0 | 84.1 | 72.0 | 94.6 | 90.1 | 89.2 | 68.5 | 76.4 | 80.2 | 80.2 | 12.6 |
| [3] (Hoyer et al., 2022a) | T | 99.1 | 74.8 | 95.2 | 61.0 | 53.1 | 59.8 | 70.7 | 68.6 | 96.0 | 63.4 | 98.7 | 84.2 | 69.9 | 95.5 | 88.9 | 84.1 | 74.0 | 70.0 | 69.7 | 77.8 | 14.2 |
| [4] (Menon et al., 2020) | T | 97.9 | 79.9 | 95.0 | 64.4 | 64.7 | 66.6 | 74.3 | 72.6 | 96.5 | 66.1 | 99.7 | 87.9 | 72.5 | 96.3 | 90.5 | 91.3 | 85.7 | 74.3 | 72.4 | 81.5 | 12.2 |
| **+BLDA** | T | 97.5 | 79.4 | 93.4 | 65.5 | 68.0 | 63.9 | 77.2 | 75.5 | 94.9 | 68.1 | 99.0 | 85.2 | 71.4 | 95.2 | 89.2 | 87.6 | 80.8 | 77.0 | 83.6 | 81.7 | 10.9 |

under-prediction classes, which achieve consistent and significant improvements. For over-prediction classes, their standard deviation (std) shows a consistent decrease, indicating more balanced predictions among these classes.

Table 18: Performance comparison on GTA.→CS. grouped by over-prediction and under-prediction classes. The results are acquired based on CNN-based model (He et al., 2016; Chen et al., 2017a), denoted as C, and Transformer-based model (Xie et al., 2021), denoted as T.

| Metrics | | IoU: mean/std | | Acc: mean/std | |
|---|---|---|---|---|---|
| Methods | Arch. | over-prediction | under-prediction | over-prediction | under-prediction |
| DACS (Tranheden et al., 2021) | C | 68.7/28.9 | 37.0/13.1 | 80.3/29.2 | 52.3/14.5 |
| **+BLDA** | C | 68.0/28.6 | 42.7/14.3 | 79.1/29.0 | 59.9/13.4 |
| DAFormer (Hoyer et al., 2022a) | C | 67.3/29.6 | 46.3/10.0 | 79.8/29.0 | 59.8/11.2 |
| **+BLDA** | C | 66.8/29.3 | 50.3/10.9 | 80.5/28.6 | 69.8/10.0 |
| CDAC (Wang et al., 2023) | T | 83.8/11.6 | 56.5/7.6 | 90.4/8.5 | 68.1/7.6 |
| **+BLDA** | T | 83.8/10.1 | 59.5/8.7 | 91.1/7.8 | 74.7/5.7 |
| DAFormer (Hoyer et al., 2022a) | T | 83.3/10.3 | 54.7/7.3 | 90.6/8.0 | 66.1/6.2 |
| **+BLDA** | T | 84.2/8.5 | 57.8/9.4 | 92.3/4.7 | 72.6/8.0 |
| HRDA (Hoyer et al., 2022b) | T | 87.8/6.8 | 61.0/8.0 | 93.0/5.6 | 72.5/10.1 |
| **+BLDA** | T | 88.6/6.2 | 64.5/7.2 | 93.9/4.0 | 77.0/8.2 |
| MIC (Hoyer et al., 2023) | T | 89.5/5.9 | 63.6/8.0 | 92.7/6.6 | 74.7/8.0 |
| **+BLDA** | T | 89.6/5.3 | 66.4/8.0 | 94.4/3.6 | 79.2/7.9 |

## M. Visualization of Estimated GMMs

In this section, we visualize the learned GMMs for target domain, i.e., $M_{cl}^T$ for all $c, l \in [C]$. Fig. 7 presents the Estimated GMMs built with DAFormer, and Fig. 8 presents the estimated GMMs with introducing BLDA. We find that the estimated GMMs can accurately model logits distribution and our method reduces the difference in logits distribution across classes, thus achieving balanced learning.

## N. More Visualization Results of Logits Distribution

In this section, we provide more visualization results to compare logits distribution built with our method. As shown in Fig.9, for over-predicted classes, the network predicts larger positive logits and negative logits (column 1, 3) , while for under-predicted classes, the network predicts smaller logits (column 2, 4). This difference in logits distribution leads to the class prediction bias. Our method reduces this difference through aligning with the anchor distribution and achieves class-balanced learning.

## O. More Qualitative Results

In this section, we provide more qualitative results between our method and other competitors on GTA→CS. As shown in Fig.10, when previous methods fail to recognize the classes that are under-predicted and suffer severe performance decline in UDA (e.g. *sidewalk*, *pole*, *fence*, *terrain*, *bike*,*sign*), BLDA shows significant improvement on them, thereby demonstrating the effectiveness of our method.

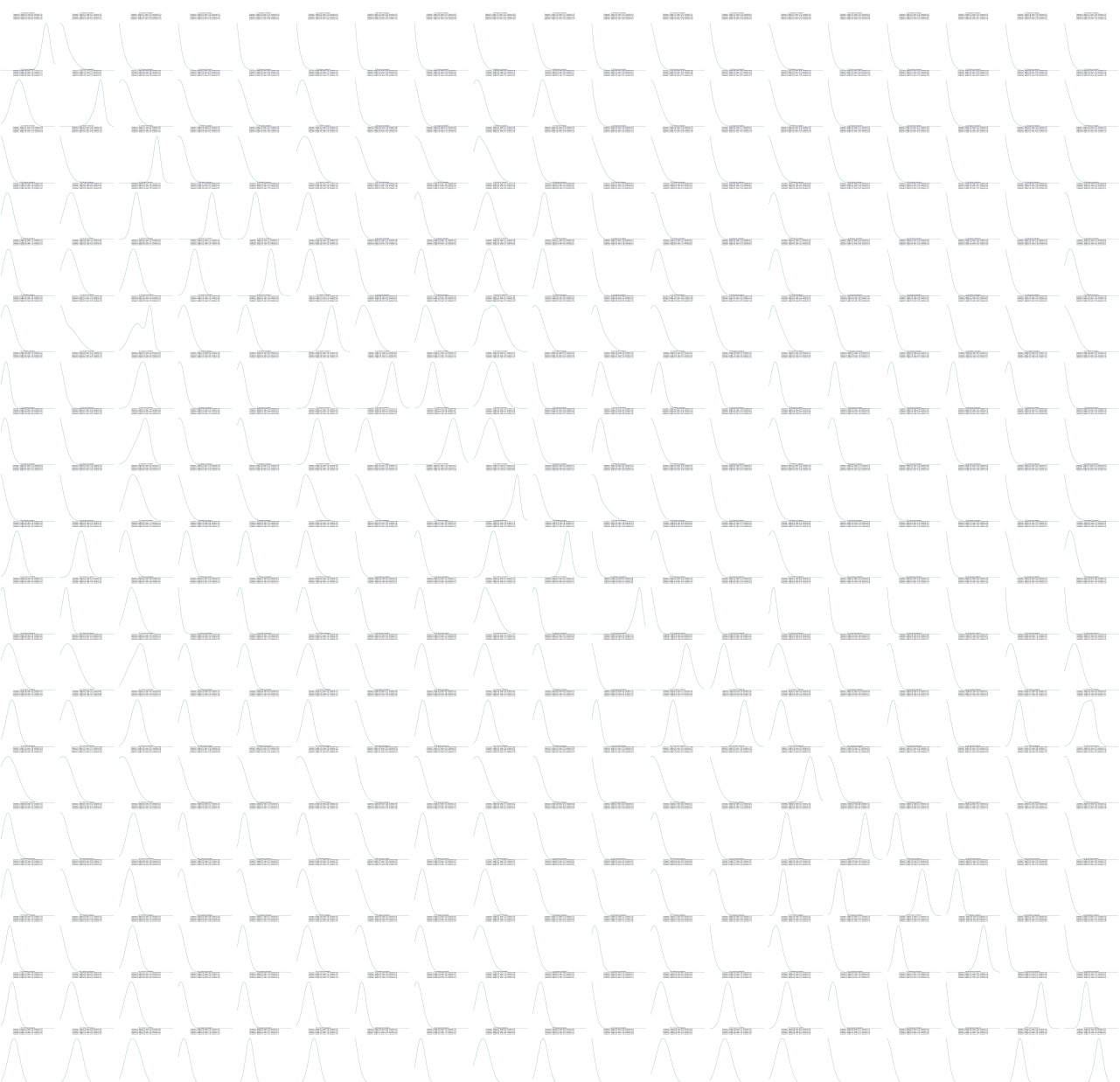

Figure 7: Estimated GMMs on target domain built with DAFormer.

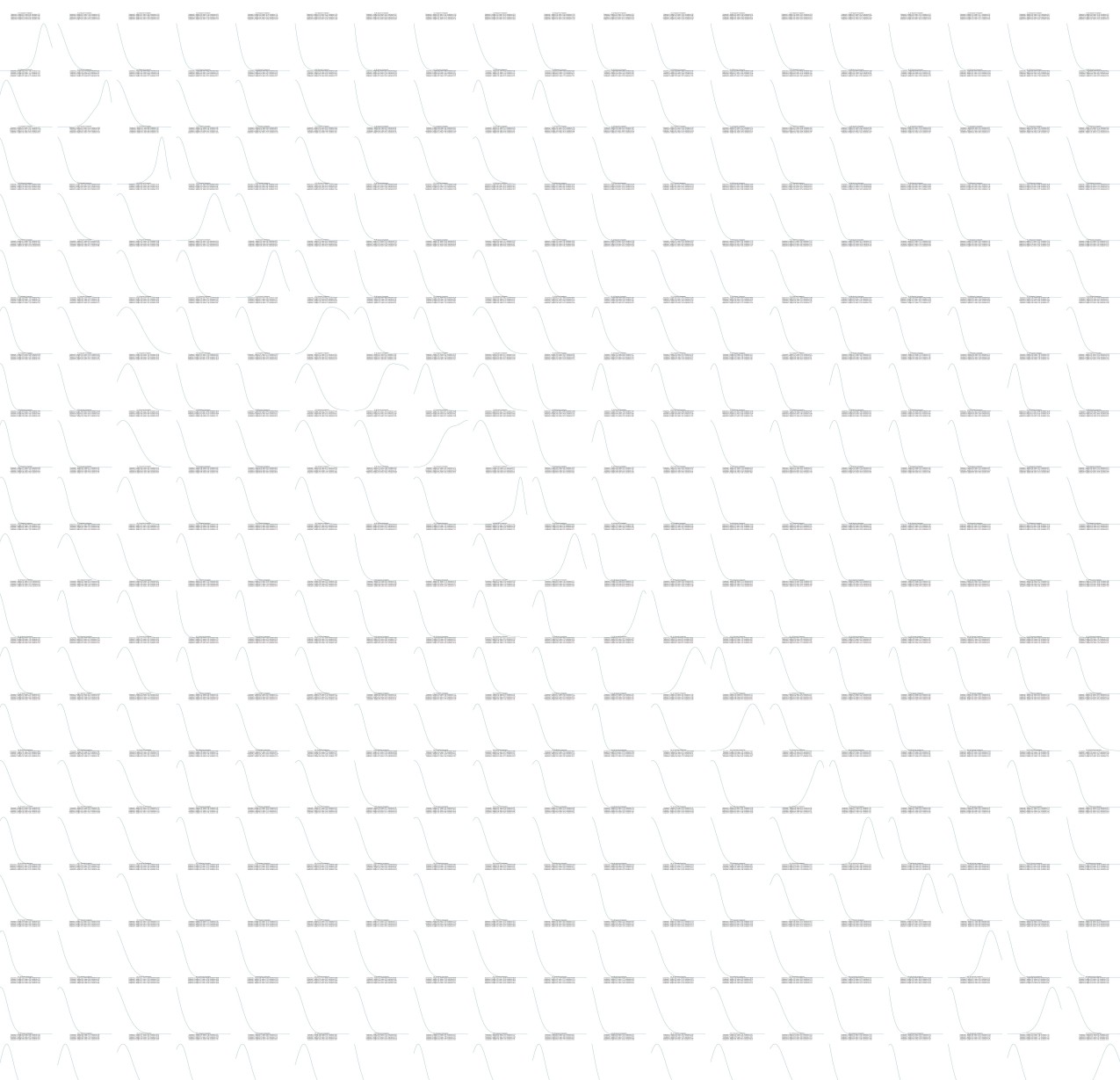

Figure 8: Estimated GMMs on target domain built with BLDA.

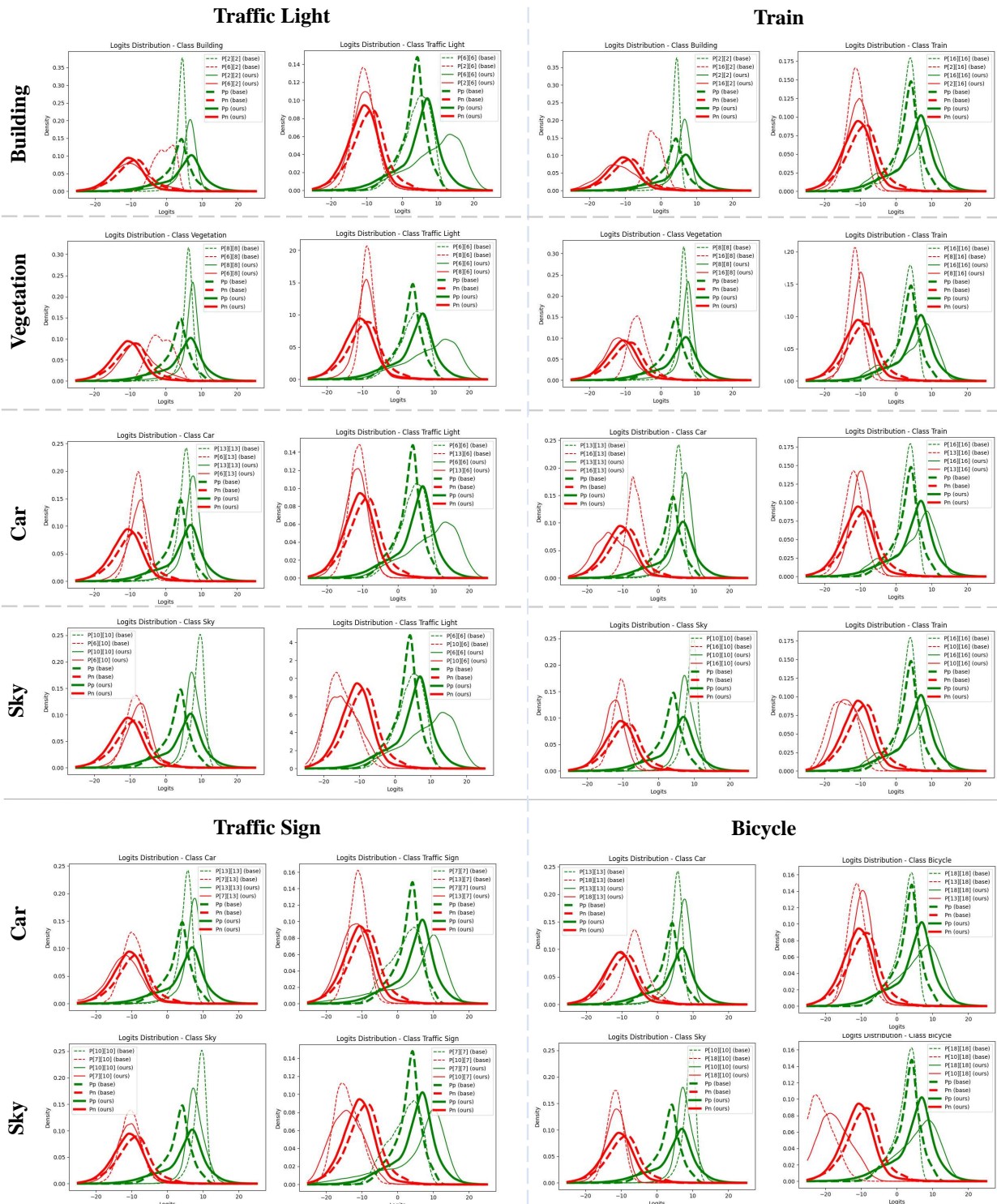

Figure 9: Comparison of Logits Distribution. We choose {*building* (2), *vegetation* (8), *car* (13) ,*sky* (10)} as over-predicted classes, and {*train* (16), *bicycle* (18), *traffic light* (6), *traffic sign* (7) } as under-predicted classes for visualization. Note that the anchor distribution is counted separately at baseline and our method.

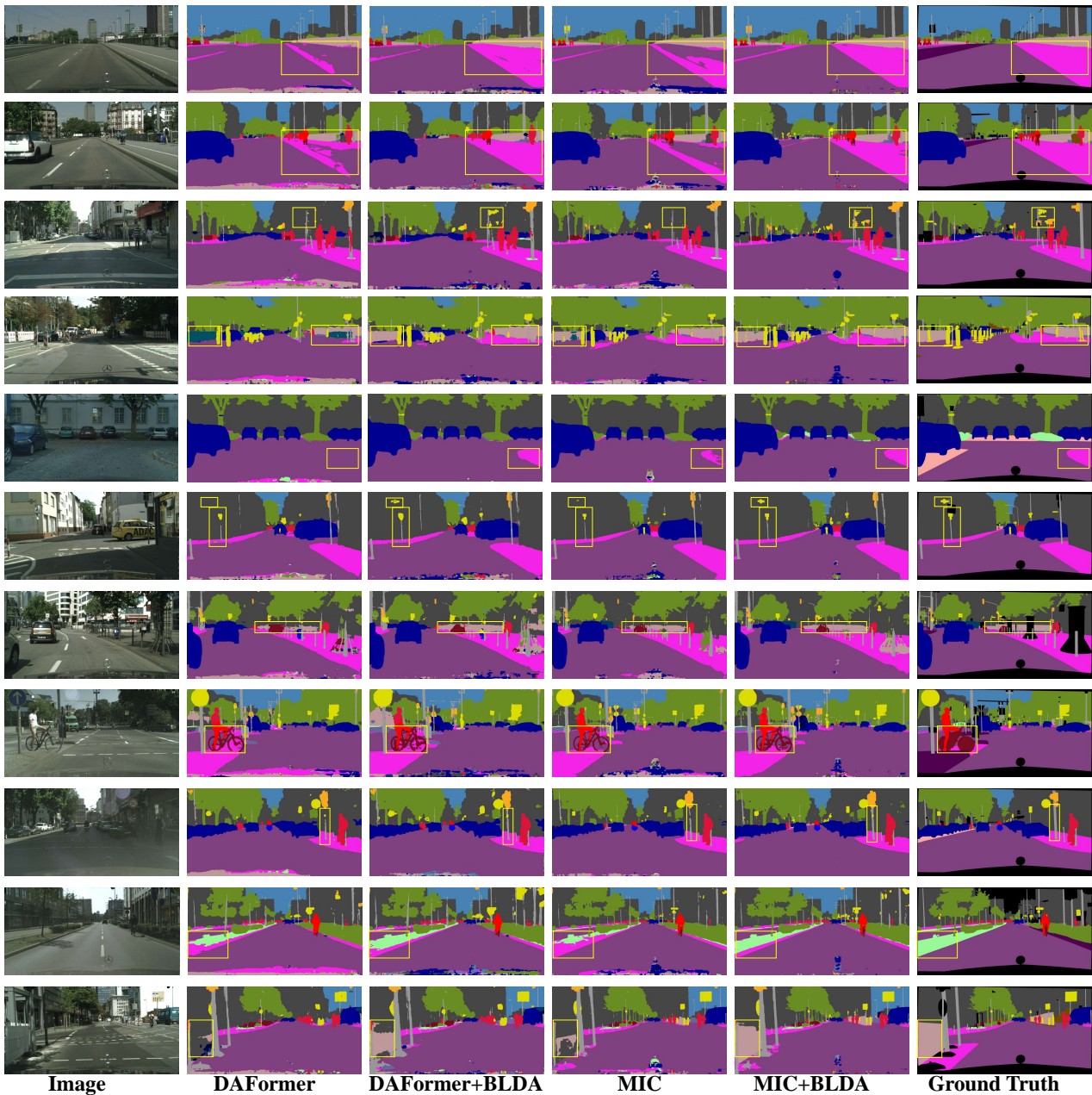

**Image**    **DAFormer**    **DAFormer+BLDA**    **MIC**    **MIC+BLDA**    **Ground Truth**

Figure 10: More qualitative comparison with DAFormer and MIC. The yellow boxes mark regions improved by BLDA.

# P. More Discussion with Related Work

Since semantic segmentation involves assigning a label to every pixel in an image, it often incurs high annotation costs (Chen et al., 2024; Li et al., 2023; Wang et al., 2024). To address this, various label-efficient approaches have been proposed, including semi-supervised learning (Sun et al., 2025; Mai et al., 2024b), few-shot learning (Li et al., 2025; Luo et al., 2024), and domain adaptation (Li et al., 2024; Chen et al., 2025). In this work, we primarily focus on the setting of unsupervised domain adaptation.

## P.1. Domain Adaptive Semantic Segmentation

Unsupervised domain adaptation (UDA) transfers semantic knowledge learned from labeled source domains to unlabeled target domains. Due to the ubiquity of domain gaps (Wangkai et al., 2023; Pan et al., 2024), UDA methods have been widely studied in various computer vision tasks, such as image classification, object detection, and semantic segmentation. UDA is crucial for semantic segmentation to avoid laborious pixel-wise annotation in new target scenarios.

Recent UDA approaches for semantic segmentation fall into two main paradigms: adversarial training-based methods (Toldo et al., 2020; Tsai et al., 2018; Chen et al., 2018; Ganin & Lempitsky, 2015; Hong et al., 2018; Long et al., 2015b) and self-training-based methods (Tranheden et al., 2021; Hoyer et al., 2022a; Araslanov & Roth, 2021; Zhang et al., 2021). Adversarial training-based methods learn domain-invariant representations through a min-max optimization game, where a feature extractor is trained to confuse a domain discriminator, aligning feature distributions across domains. Self-training-based methods, which have become dominant in the field due to the domain-robustness of Transformers (Bhojanapalli et al., 2021), generate pseudo labels for target images based on a teacher-student optimization framework. The success of this paradigm depends on generating high-quality pseudo labels, with strategies such as entropy minimization (Chen et al., 2019) and consistency regularization (Hoyer et al., 2023) being developed for this purpose.

Recently, several approaches have been proposed to tackle the challenges in UDA for semantic segmentation from different perspective: DTS (Huo et al., 2023) employs a Dual Teacher-Student Framework, promoting the self-training paradigm to fully adapt models to the target domain by exploring different mix strategies. CDAC (Wang et al., 2023) introduces consistency constraints in attention to enhance the model's cross-domain performance. RTea (Zhao et al., 2023) defines proxy tasks based on structural information and incorporates them into the self-training paradigm as additional supervision signals. Peng et al. (2023) applies Diffusion-based image translation techniques to directly mitigate the distribution differences between the target and source domains. DiGA (Shen et al., 2023) integrates distillation strategies and self-training through multi-stage training. Copt (Mata et al., 2025) uses domain-agnostic text embeddings to learn domain-invariant features in an image segmentation encoder. MICDrop (Yang et al., 2025) learns a joint feature representation by masking image encoder features while inversely masking depth encoder features to leverage geometric information. ADFormer (He & Todorovic, 2025) introduces a new transformer by decomposing cross-attention in the decoder into domain-independent and domain-specific parts

However, due to the inherent class imbalance and distribution shift in both data and label space between domains, networks often exhibit complex class biases, which are further amplified by the confirmation bias inherent in the self-training paradigm. Our method aims to achieve balanced learning in UDA training to mitigate these issues and improve the overall performance of domain adaptation for semantic segmentation.

## P.2. Class-Imbalanced Learning

Class imbalance is a prevalent issue in semantic segmentation, where the number of samples per class varies significantly. Existing methods tackle this problem through re-weighting or re-sampling techniques. Re-weighting methods assign different weights to classes during training, giving higher importance to under-represented classes (Cui et al., 2019; Lin et al., 2017; Cao et al., 2019; Liu et al., 2019). Re-sampling techniques modify the class distribution in the training data by over-sampling minority classes or under-sampling majority classes (He et al., 2008; 2021; He & Garcia, 2009; Kim et al., 2020; Chu et al., 2020).

Recent works have proposed various approaches to address class imbalance in different tasks. For object detection, Lin et al. (2017) propose a re-weighting method that assigns weights to samples based on their confidence, with harder samples receiving higher weights. In classification, Cui et al. (2019) define an effective number of samples for each class to weight different classes, while Menon et al. (2020) introduce a logit adjustment method that incorporates correction terms in logits, determined by prior knowledge of class distribution. For segmentation, Chu et al. (2021) propose a stochastic training

scheme for semantic segmentation, which improves the learning of debiased and disentangled representations, and Wang et al. (2021a) propose a Seesaw loss that reweights the contributions of gradients produced by positive and negative instances of a class for instance segmentation. Sun et al. (2025) introduces a generative approach that alleviates bias by modeling the task as conditional discrete data generation with a mathematically derived debiasing strategy.

In UDA for semantic segmentation, several approaches have introduced these strategies to alleviate class bias: DAFormer (Hoyer et al., 2022a) present a re-sampling technique that samples based on prior knowledge of class distribution, with more sampling for rare classes. Freedom (Truong et al., 2023) considers class-imbalance as fairness problem and propose to model the context of structural dependency to tackle it. SAC (Araslanov & Roth, 2021) cit maintains an exponentially moving class prior used to discount the confidence thresholds. CPSL (Li et al., 2022) employs a distribution alignment technique to enforce the marginal class distribution of cluster assignments to be close to that of pseudo labels. However, these methods are still empirical and focus on the single-domain setting, which assumes that the test data and training data share the same distribution in both data space and label space, without considering the additional challenges posed by domain shift in UDA.

In contrast to these methods, our approach aims to directly address class bias and achieve balanced learning for each class without relying on prior knowledge about the distribution shift between domains. We evaluate class bias through logit sets in the form of a confusion matrix and explicitly balance components in the matrix using anchor distributions and cumulative density functions, implemented in an online self-training paradigm. Our method does not depend on prior knowledge of class distribution and instead directly models class bias based on the actual logit distribution, making it more adaptable to the challenges posed by domain shift in UDA.

### P.3. Further Comparison with Existing Methods

While our method shares some similarities with existing works (Zhang et al., 2019; Ning et al., 2021; Wang et al., 2021b; Moradinasab et al., 2024) in terms of introducing anchor distributions for alignment and using GMMs for distribution estimation, our approach fundamentally differs in its goal of addressing the unique challenge of class bias in UDA settings. Specifically:

Previous methods focus primarily on mitigating domain discrepancy through feature alignment strategies. For example, Zhang et al. (Zhang et al., 2019) use class-wise anchors to represent source domain distributions and align target domain features accordingly. Ning et al. (Ning et al., 2021) employ multiple anchors to model potentially multimodal category distributions and select target samples with high discrepancy for active learning. Wang et al. (Wang et al., 2021b) leverage contrastive learning on target domain features using prototypes modeled by GMMs on the source domain. Moradinasab et al. (Moradinasab et al., 2024) identify reliable pseudo-labels through uncertainty modeling with GMMs.

In contrast, our method proposes a class balanced learning strategy targeting class bias, rather than the feature alignment strategies employed by previous methods to address domain differences. We propose a complete methodology (*why–what–how*) specifically for UDA settings. We first analyze the unique class bias challenge, distinct from regular class-imbalanced problems (*why*). Then, based on this unique challenge, we carefully design a class balanced learning strategy to estimate correction terms from a logits distribution perspective (*what*). Finally, to implement this strategy, we first implement distribution estimation as a prerequisite, then prepend to delicate online logits adjustment to perceive the model's learning state, towards class-balanced learning (*how*).

## Q. Limitation

Our method analyzes the class bias in domain adaptive semantic segmentation through logits distribution statistics and propose a method to implement online logits adjustment tailored for the UDA training process, which can be easily built with exiting methods and demonstrate consistent improvements. Although BLDA achieves remarkable performance, we balance the class under the assumption that each logits distribution in $\mathcal{M}$ is independent, without considering correlation between classes. How to model this correlation and mitigate the class bias further is still to be resolved.

