# OpenReview forum: "Balanced Learning for Domain Adaptive  Semantic Segmentation"
_ICML.cc/2025/Conference — ICML 2025 poster_

### Official Review · Reviewer_egQB · 2025-03-13

**Overall Recommendation:** 3

**Summary:**

The paper proposes Balanced Learning for Domain Adaptation (BLDA) to address class bias in unsupervised domain adaptation (UDA) for semantic segmentation. BLDA analyzes logits distributions to assess prediction bias and introduces an online logits adjustment mechanism to balance the class learning in both source and target domains. Experimental results demonstrate consistent performance improvements when integrating BLDA with various methods on standard UDA benchmarks.

**Claims And Evidence:**

The claims made in the submission are supported by clear and convincing evidence.

**Essential References Not Discussed:**

No related works that are essential to understanding the context for key contributions of the paper, but are not currently cited/discussed in the paper.

**Experimental Designs Or Analyses:**

I have checked the experimental results. What are the experimental results of unsupervised domain adaptation at Cityscapes to ACDC[1]？
[1] ACDC: The Adverse Conditions Dataset with Correspondences for Semantic Driving Scene Understanding. ICCV 2021

**Methods And Evaluation Criteria:**

The proposed method makes sense for the problem and experiment results demonstrate its effectiveness.

**Other Comments Or Suggestions:**

No

**Other Strengths And Weaknesses:**

The writing is excellent, the charts are beautiful, and the formulas are exciting!
The experiments are slightly lacking and would like to show more experimental results for unsupervised domain adaptation settings such as Cityscapes to ACDC.

**Questions For Authors:**

What are the experimental results of unsupervised domain adaptation at Cityscapes to ACDC[1]？
[1] ACDC: The Adverse Conditions Dataset with Correspondences for Semantic Driving Scene Understanding. ICCV 2021

**Relation To Broader Scientific Literature:**

BLDA analyzes logits distributions to assess prediction bias and introduces an online logits adjustment mechanism to balance class learning in both source and target domains. It inspires more researchers to utilize logit distributions to assess prediction bias.

**Theoretical Claims:**

I have checked the correctness of any proofs for theoretical claims. The paper states that "the distribution of logits predicted by the network can assess the degree of class bias", which is demonstrated by Fig.1(c) and Fig.1(d).

---

> ### Author Rebuttal · Authors · 2025-03-31
>
> We sincerely thank the reviewer for the positive and constructive feedback. We appreciate your recognition of  **motivation**, **theoretical intuition**, and  **clear writing**, as well as your kind comments on our visualizations and formula design.
>
> ---
>
> Regarding your suggestion to include more experimental results for the Cityscapes → ACDC setting, we agree that evaluating BLDA under more diverse and challenging UDA scenarios is important and we conduct extended experiments on image classification/video semantic segmentation in Appendix G/H. We conduct additional experiments on the Cityscapes → ACDC benchmark and present the results below.
>
> As shown in the following tables (∗ denotes the reproduced result), integrating BLDA consistently improves performance across multiple baselines, including DACS, DAFormer, and MIC, on both mIoU and mAcc metrics. Importantly, BLDA is designed as a plug-and-play module that is model-agnostic and can be seamlessly integrated into most existing UDA baselines without modifying their core architectures or training objectives. Since class bias is a pervasive issue across UDA settings, our method provides a general mechanism to mitigate this imbalance during training. As such, BLDA is able to consistently improve performance across a wide range of models and target domains, as demonstrated in our experiments on both standard benchmarks and the newly added Cityscapes → ACDC setting.
>
> **Cityscapes → ACDC (IoU, %)**
>
> | Method| Arch. |Road|Sidewalk|Building|Wall|Fence|Pole|Light | Sign  | Veg | Terrain  | Sky  | Person | Rider    | Car  | Truck    | Bus      | Train    | Motor    | Bike     | **mIoU** | **std**  |
> |-|-|-|-|-|-|-|-|-|-|- |- |- |-|- |- |- | - | - |- | - | - |- |
> | DACS* | C | 78.5 | 36.3  | 73.4 | 28.7 | 17.0  | 42.2 | 57.1  | 45.6  | 68.3 | 26.7 | 75.0| 51.3 | 24.5  | 75.6 | 37.8 | 43.1 | 41.3 | 28.8     | 26.8     | 46.0     | 19.3     |
> | **+BLDA** | C  | 74.6 | **41.9** | 67.3     | **31.5** | **19.8** | **46.9** | 54.8  | **50.9** | **71.7** | **29.5** | 69.4     | **54.6** | **26.6** | 74.2 | **41.9** | **45.0** | **49.8** | **31.1** | **27.9** | **47.9** | **17.1** |
> | DAFormer* | T   | 65.5 | 52.8     | 79.1     | 39.8     | 37.2 | 56.4 | 57.6  | 51.3  | 72.2| 37.3| 59.9     | 54.7     | 25.0     | 83.6 | 69.6     | 68.0     | 72.9     | 39.6     | 33.2     | 55.5     | 16.3     |
> | **+BLDA** | T  | 63.8 | 50.7     | 73.4     | **40.2** | **40.7** | 53.3     | 54.6  | 50.0     | 70.7     | **40.3** | **64.3** | **58.9** | **33.9** | 83.5 | **74.3** | **75.5** | **78.9** | **45.1** | **37.6** | **57.4** | **15.2** |
> | MIC*  | T     | 89.5 | 63.0 | 86.3 | 56.7 | 46.2     | 64.2     | 65.8  | 64.2     | 75.5     | 46.9     | 83.2     | 67.9     | 45.6     | 87.7 | 85.9     | 92.3   | 88.7| 54.0| 55.4   | 69.4 | 15.9 |
> | **+BLDA** | T     | 88.7 | **66.8** | **87.8** | **61.1** | **48.9** | 63.3     | 65.2  | **67.3** | 73.7     | **50.5** | 82.4     | **68.2** | **49.5** | 86.7 | 80.5  | 91.1 | **91.0** | **54.6** | **61.8** | **70.5** | **14.2** |
>
> **Cityscapes → ACDC (Acc, %)**
>
> | Method    | Arch. | Road  | Sidewalk | Building | Wall     | Fence    | Pole     | Light    | Sign  | Veg  | Terrain  | Sky | Person   | Rider    | Car      | Truck    | Bus      | Train    | Motor    | Bike     | **mAcc** | **std**  |
> | -- | - | -| --| - | - | -| -- | - | - | - | -- | --- | --- | --- | --- | -- | -- | ---- |--- | ---- | --- | - |
> | DACS*     | C     | 96.7     | 52.9     | 81.9     | 39.2     | 36.3     | 51.8     | 73.1     | 55.0     | 90.7 | 35.0     | 76.1     | 60.1     | 33.3     | 84.5     | 44.3     | 44.2     | 59.7     | 35.8     | 39.8     | 57.4     | 20.0     |
> | **+BLDA** | C     | 96.1     | **66.7** | **82.7** | **40.7** | **36.6** | **58.9** | 66.4     | **67.4** | 84.1 | **41.3** | 70.7     | **68.4** | **36.3** | **84.8** | **48.9** | **46.5** | **59.9** | **47.4** | **41.4** | **60.3** | **17.7** |
> | DAFormer* | T  | 98.2| 63.4  | 86.7  | 51.2| 42.0  | 70.3     | 77.3     | 65.1     | 93.4 | 52.6     | 60.6     | 60.3     | 53.7     | 90.7     | 89.3 | 70.8  | 90.3 | 57.8| 42.7 | 69.3  | 17.4 |
> | **+BLDA** | T | **98.8** | 61.3     | 85.5   | **55.8** | **45.4** | **73.2** | **82.5** | **72.8** | 92.0 | 51.7     | **65.0** | **69.2** | 51.5 | 90.5| **89.5** | **80.0** | **92.8** | 60.6 | **51.1** | **72.1** | **16.4** |
> | MIC* | T  | 99.4  | 69.2  | 92.2 | 76.0 | 57.7| 73.5 | 89.5     | 80.9     | 94.9 | 62.8  | 89.3 | 82.7     | 57.5     | 97.6     | 91.9  | 96.9     | 96.6     | 60.2     | 72.6     | 81.1     | 14.2     |
> | **+BLDA** | T  | 98.8     | **72.3** | **93.8** | **86.1** | **59.7** | **73.6** | **90.8** | **89.9** | 93.2 | **73.9** | **90.8** | 81.4     | **68.3** | 96.3 | **94.8** | **97.0** | **97.6** | **65.3** | 74.1 | **84.1** | **12.1** |
>
> We will include these results in the revised version of the paper.
>
> ---
>
> We hope our response can resolve your concern. Please do not hesitate to let us know if you have further questions.

---

### Official Review · Reviewer_41Vm · 2025-03-13

**Overall Recommendation:** 3

**Summary:**

This paper presents Balanced Learning for Domain Adaptation (BLDA), an innovative approach to address class imbalance and distribution shifts in Unsupervised Domain Adaptation (UDA) for semantic segmentation. Specifically, it identifies over-predicted and under-predicted classes through the analysis of predicted logits and employs a post-hoc approach to align logits distributions across different classes using shared anchor distributions. During self-training, BLDA estimates logits distributions online and incorporates correction terms into the loss function to ensure unbiased pseudo-label generation. Extensive experiments on standard UDA benchmarks have demonstrated that BLDA consistently improves performance, particularly for under-predicted classes, when integrated with various existing methods.

**Claims And Evidence:**

Yes

**Essential References Not Discussed:**

N/A

**Experimental Designs Or Analyses:**

This paper has constructed extensive experiments on the standard UDA benchmarks to demonstrate the effectiveness of the proposed post-hoc method.

**Methods And Evaluation Criteria:**

Yes

**Other Comments Or Suggestions:**

N/A

**Other Strengths And Weaknesses:**

Strengths:

1. The paper is generally well-written, well-structured, and easy to follow.
2. The experiments are comprehensive, covering three transfer tasks for segmentation, an additional image classification task (included in the supplementary materials), and extensive qualitative analyses.

Weaknesses:
1. The paper claims to stress the class-imbalanced problem by the proposed BLDA. However, it seems like the proposed method profoundly depends on the baseline models. For example, the performance of "Train" in Table 1 is close to 0 for DACS and DAFormer (C), which shows that the proposed method may still suffer from the class-imblanced problem.
2.  Concerns regarding the fairness of the comparisons. In Tables 1 and 2, the experiments seem to leverage high-quality pseudo-labels for self-training, which may inherently provide an advantage over existing methods, such as DAFormer, CDAC, HRDA, and MIC. This raises questions about whether the improved performance is due to the proposed method's unique contributions or simply a result of the enhanced quality of the pseudo-labels used.
3. It would be more appropriate to compare the performance with the existing method beginning with the same source-only model.

**Questions For Authors:**

The major question is about Eq. 4. As $\mathbb{P}(\arg max_{c' \in [C]}f_{\theta}(x)[c']=l\|y=c)$ represents the probability of predicting class $c$ as $l$ and $c \neq l$, why does the positive bias $\mathrm{Bias}(l)$ indicates over-prediction? Given the condition $y=c$, how do we explain the representation of the summation of $c' \in [C]$? As my understanding, it may be $\mathbb{P}(\arg max_{l \in [C]}f_{\theta}(x)=l\|y=c)$.

**Relation To Broader Scientific Literature:**

The class imbalance problem is widely studied in semantic segmentation, cross-domain semantic segmentation, and other perception fields. This paper proposes a post-hoc method to balance over-prediction and under-prediction classes during domain adaptation, which is straightforward and makes sense.  Moreover, the proposed method may cost a lot of time and computational resources due to the multiple GMMs.

**Theoretical Claims:**

There is some confusion about Eq.(4).  As $\mathbb{P}(\arg max_{c' \in [C]}f_{\theta}(x)[c']=l\|y=c)$ represents the probability of predicting class $c$ as $l$ and $c \neq l$, why does the positive bias $\mathrm{Bias}(l)$ indicates over-prediction? Given the condition $y=c$, how do we explain the representation of the summation of $c' \in [C]$? As my understanding, it may be $\mathbb{P}(\arg max_{l \in [C]}f_{\theta}(x)=l\|y=c)$.

---

> ### Author Rebuttal · Authors · 2025-03-31
>
> We sincerely thank the reviewers for their valuable feedback and thoughtful comments. We appreciate the recognition of our **clear writing**, **comprehensive experiments**, and **extensive qualitative analyses**. We address each of your concerns point by point.
>
> ---
>
> **Q1: Clarification on Eq. (4) and the Definition of Positive Bias**
>
> **A1:**  Sorry for any misunderstanding. In Eq. (4), the expression $\arg\max_{c' \in [C]} f_\theta(x)[c']$ refers to the predicted class for input $x$.  $P(\arg\max_{c' \in [C]} f_\theta(x)[c'] = l \mid y = c)$  represents the probability that a sample from class $c$ is predicted as class $l$. There is no constraint that $c \ne l$ in Eq. (4). The summation in this equation is taken over the conditioning variable $c$, i.e., it averages the conditional probabilities $P(\arg\max_{c' \in [C]} f_\theta(x)[c'] = l \mid y = c)$ across all classes $c \in [C]$.
>
> By summing over all $c$ and taking the average, we obtain the expected probability that a sample from any class (including $l$ itself) is predicted as class $l$. Under an unbiased model, this expectation should be approximately $1/C$. Therefore, a positive bias indicates that class $l$ is over-predicted, as its average prediction probability exceeds the uniform expectation.
>
> ---
>
> **Q2: Dependence on Baseline Models**
>
> **A2:** Our method is designed as a plug-and-play module that can be integrated into any self-training-based UDA framework. Naturally, its performance is influenced by the underlying baseline, especially when estimating the target-domain logits distribution, which relies on the pseudo-labels generated during self-training. Hence, the quality of pseudo-labels affects the accuracy of our distribution estimation and, consequently, the effectiveness of the class-aware adjustment.
>
> However, we emphasize that BLDA consistently reduces class-level prediction variance across all baselines. As reported in the main paper, the standard deviation of per-class IoU and Acc drops significantly when BLDA is applied. This demonstrates that our method consistently yields more balanced predictions with reduced class bias, regardless of the baseline.
>
> ---
>
> **Q3: Fairness of Comparisons and Pseudo-Label Quality**
>
> **A3:** We would like to clarify that our method adheres to the standard UDA setting and does not modify the pseudo-label generation process of any baseline. As described in Section 3.2, we employ the self-training framework as implemented in existing works such as DACS, DAFormer, CDAC, HRDA, and MIC. All these methods adopt an online self-training protocol: the model is trained from scratch using labeled source data and unlabeled target data with pseudo-labels generated during training.
>
> BLDA is inserted into this process as a lightweight module that adjusts the logits distributions based on estimated class-wise prediction behavior. The pseudo-label generation and training pipeline of each baseline remains unchanged.
>  Therefore, the comparisons are conducted under fair and consistent settings,  where each method follows the same online self-training paradigm (note that all baselines train the model from scratch, without using a source-only pretrained model).
>
>
> ---
>
> **Q4: Computation Overhead**
>
> **A4:** We provide a detailed analysis of the computational overhead in Appendix I, including actual resource usage and training time across various baselines. In summary, the additional cost introduced by our proposed components stems from three main operations. We have implemented them efficiently to minimize overhead:
>
> 1. **GMM Implementation**: Instead of using off-the-shelf libraries that update Gaussian parameters sequentially, we store the parameters of all $C \times C \times K$ components as tensors in PyTorch and update them in parallel using matrix operations.
> 2. **CDF Computation**: We approximate the cumulative distribution function using the Abramowitz-Stegun formula, which allows efficient polynomial evaluation.
> 3. **Inverse CDF Computation**: We use interpolation techniques within the estimated value range, which avoids costly numerical inversion.
>
> All the above operations can be efficiently performed using simple matrix operations on tensors. Moreover, the storage of Gaussian component parameters and the additional regression head (a $1\times 1$ conv) introduced by our method are lightweight. Overall, our method demonstrates high efficiency in both training time and GPU memory, as reported in Table 13 of Appendix I.  Moreover, our method introduces no additional overhead during inference.
>
> ---
>
> We hope our response can resolve your concern. Please do not hesitate to let us know if you have further questions.

---

### Official Review · Reviewer_j9eK · 2025-03-14

**Overall Recommendation:** 4

**Summary:**

The paper proposes Balanced Learning Domain Adaptation for addressing class bias in unsupervised domain adaptative semantic segmentation. The authors identify class imbalance and distribution shifts as major obstacles in UDA and propose techniques to analyze logit distributions to assess prediction bias. The method introduces post-hoc logits adjustment and online logits adjustment to mitigate class bias and improve balanced learning across classes. Additionally, cumulative distribution estimation is used as domain-shared structural knowledge. The paper demonstrates significant performance improvements on standard UDA benchmarks.

**Claims And Evidence:**

1. The claim that logit distribution differences correlate with class bias is well-supported by experimental visualizations of FIG 6.
2. The effectiveness of online logits adjustment and post-hoc adjustment is supported by ablation studies。
3. The effectiveness of Balanced Learning is supported by consistent improvements in mIoU and mAcc metrics across several baselines and benchmarks.

**Essential References Not Discussed:**

None

**Experimental Designs Or Analyses:**

The experimental designs are sound, with appropriate baselines and thorough ablation studies.
The use of mIoU and mAcc metrics is effective for assessing both overall accuracy and balanced performance across classes.

**Methods And Evaluation Criteria:**

1. The proposed methods, including post-hoc logits adjustment and online logits adjustment, are well-suited to address the stated problem of class bias in UDA.
2. The use of standard UDA benchmarks ensures fair and meaningful evaluation.

**Other Comments Or Suggestions:**

1. The scalability of the method should be discussed, especially in terms of computational overhead introduced by GMMs.

**Other Strengths And Weaknesses:**

Strengths:
1. The method is versatile and can be integrated with various self-training-based UDA frameworks.
2. The authors provide comprehensive experiments, including multiple benchmarks and ablation studies.
3. The paper provides a clear theoretical explanation for logit distribution alignment and its connection to class bias, grounding the proposed method in established statistical principles.

Weakness:
1. The GMM-based logits modeling may be computationally expensive, especially for larger datasets.
2. The reliance on pre-defined anchor distributions may not generalize well to all datasets or domain shifts. The paper lacks discussion on alternative approaches, such as learned or adaptive anchors.

**Questions For Authors:**

1. How does the method scale to larger datasets or real-time applications, given the computational cost of GMM-based logits modeling?

**Relation To Broader Scientific Literature:**

The idea of using logits distribution analysis is related to class-imbalanced learning and focusing on distribution shifts between domains.

**Theoretical Claims:**

No major theoretical claims were presented beyond the statistical modeling of logits distributions and their alignment.

---

> ### Author Rebuttal · Authors · 2025-03-31
>
> We sincerely thank the reviewer for the positive and constructive feedback. We appreciate your recognition of **our method’s versatile   design**, the **thoroughness of our experimental validation**, and the **clarity of our theoretical explanation**. We address each of your concerns point by point.
>
> ---
>
>
> **Q1: Computational Cost and Scalability of GMM-based Modeling**
>
> **A1:**  We provide a detailed analysis of the computational overhead in Appendix I, including actual resource usage and training time across various baselines. In summary, the additional cost introduced by our proposed components stems from three main operations. We have implemented them efficiently to minimize overhead:
>
> 1. **GMM Implementation**: Instead of using off-the-shelf libraries that update Gaussian parameters sequentially, we store the parameters of all $C \times C \times K$ components as tensors in PyTorch and update them in parallel using matrix operations.
> 2. **CDF Computation**: We approximate the cumulative distribution function using the Abramowitz-Stegun formula, which allows efficient polynomial evaluation.
> 3. **Inverse CDF Computation**: We use interpolation techniques within the estimated value range, which avoids costly numerical inversion.
>
> All the above operations can be efficiently performed using simple matrix operations on tensors. Moreover, the storage of Gaussian component parameters and the additional regression head (a $1\times 1$ conv) introduced by our method are lightweight. Overall, our method demonstrates high efficiency in both training time and GPU memory, as reported in Table 13 of Appendix I.  Moreover, our method introduces no additional overhead during inference.
>
> For larger datasets, i.e., with more classes, the only change is that the GMM module needs to store a proportionally larger number of Gaussian components, which remains tractable in practice.
>
> ---
>
> **Q2: Generalization of Pre-defined Anchor Distributions**
>
> **A2:** We provide a detailed discussion of the anchor distribution design and its alternatives in Appendix J. A summary is provided below:
>
> 1. **Definition and Role of Anchor Distributions:**
>     We use the global positive and negative logits distributions from the source domain to estimate a shared anchor distribution for both the source and target domains. This shared anchor allows the target logits distribution to gradually align with the source, serving two purposes:
>
>    - It provides a reference to balance learning progress across classes within each domain.
>    - It acts as a bridge to align the source and target domains in terms of class-wise logits behavior.
>
> 2. **Different Selection Criteria for Anchor Distributions:**
> While the anchor distribution is estimated from the source domain, we acknowledge that some discrepancy may exist between this estimation and the true distribution of the target domain. However, our analysis in Appendix B shows that the relative pos/neg bias is more important than the absolute value of the logits, and such relative structure tends to be preserved across domains due to the observation that the biases of positive and negative logits are coupled in Fig.1 (c). This explains why the impact of distributional mismatch is limited in practice.
>
>       We also conduct ablation studies (Table 14) on different anchor selection strategies, including: Global source distribution (default); Global target distribution (oracle); Two biased classes (*building* and *fence* in Fig. 1(d)). The results show that different anchor choices lead to only marginal differences in performance, further demonstrating the robustness of our method to anchor selection and its ability to generalize across domain shifts.
>
> 3. **Generalization Across Datasets and Domain Shifts:**
>     Our logits adjustment mechanism can be interpreted as a cross-entropy loss with an adaptive margin. As discussed in Appendix B, this margin is implicitly determined by the relative difference between the positive and negative logits distributions. There are two key cases:
>    - If both positive and negative logits in the anchor increase or decrease simultaneously, the margin remains stable.
>    - If the relative gap between pos/neg distributions in the target differs from that in the anchor, the margin adapts accordingly to guide alignment.
>
>    This behavior enables the method to generalize across domain shifts. As demonstrated in Table 15,   the adaptive margin mechanism drives the target logits distribution to progressively align with the anchor distribution over training, confirming the intended behavior of our method.
>
> ---
>
> We hope our response can resolve your concern. Please do not hesitate to let us know if you have further questions.

---

### Official Review · Reviewer_ZPAk · 2025-03-17

**Overall Recommendation:** 3

**Summary:**

The study conducts an in-depth investigation into class bias within UDA scenarios, demonstrating that this bias stems from simultaneous shifts in both the label and data distributions, which complicates the domain adaptation process. To address this challenge, the authors introduce a novel approach that evaluates and reduces class bias through a class-balanced learning method. This approach derives adjustment factors from the distribution of logits, thereby overcoming the constraints inherent in conventional imbalanced class techniques. Notably, the proposed solution is implemented as a versatile plug-and-play module, suitable for broad applications in UDA. It begins by estimating distribution parameters, then applies dynamic logit adjustments to monitor the model’s learning progression, ensuring balanced class performance. Extensive experiments confirm that the method consistently yields significant improvements in performance, highlighting its effectiveness and flexibility.

## update after rebuttal
Thanks for the authors' detailed response. The response has addressed most of my concerns. I hope the authors can revise their paper according to the reviewers' suggestions. I will keep my original rating.

**Claims And Evidence:**

The experimental claims are supported with thorough empirical results. However, my concern is the authors need further justification about why using logits distributions as a proxy for class imbalance. Additionally, the method needs comparison against alternative distribution estimation methods except using Gaussian Mixture Models.

**Essential References Not Discussed:**

There are plenty of domain adaptive semantic segmentation methods. I know it is not possible for the authors to plug their method to each of them to show the effectiveness of their method, but some typical methods representing different technical solutions need to be compared. For example, comparing to the works listed below

[1] Learning to Adapt Structured Output Space for Semantic Segmentation, CVPR 2018.

[2] Confidence regularized self-training, ICCV 2019.

[3] Pixel-Level Cycle Association: A New Perspective for Domain Adaptive Semantic Segmentation, NeurIPS 2020.

**Experimental Designs Or Analyses:**

The experimental systematically demonstrating improvements introduced by BLDA. The authors conduct extensive ablations on each model and various configurations, which strongly support the design.

**Methods And Evaluation Criteria:**

The selected metrics—mIoU and mAcc are well-suited for assessing performance, as they capture both the precision of segment overlap and overall classification accuracy. Additionally, the method itself is thoughtfully designed, with a clear and sound intuition underpinning its architecture and approach.

**Other Comments Or Suggestions:**

No other comments.

**Other Strengths And Weaknesses:**

Strengths:

-Clear written paper.

-Clearly identified problem of class imbalance unique to UDA.

Weaknesses:

-Lack of analysis concerning computational overhead and scalability.

-Need further justification about why using logits distributions as a proxy for class imbalance.

-Need further comparison against alternative distribution estimation methods except using Gaussian Mixture Models.

**Questions For Authors:**

Please refer to Other Strengths And Weaknesses.

**Relation To Broader Scientific Literature:**

The work is well related with domain-adaptive semantic segmentation literature, clearly point out the limitations of prior class-balancing methods such as re-weighting and re-sampling.

**Theoretical Claims:**

The theoretical claims about the connection between logits distribution and class biases are well-defined. The authors clearly depict their relationships and provide convincing theoretical justifications.

---

> ### Author Rebuttal · Authors · 2025-03-31
>
> We thank the reviewer for the positive and constructive feedback, as well as for acknowledging our contributions,  including the **clear problem formulation**, the **clarity of the writing**, and the **effectiveness of  empirical validation**. We address each of your concerns point by point.
>
> ---
>
> **Q1: Computation Overhead**
>
> **A1:** We provide a detailed analysis of the computational overhead in Appendix I, including actual resource usage and training time across various baselines. In summary, the additional cost introduced by our proposed components stems from three main operations. We have implemented them efficiently to minimize overhead:
>
> 1. **GMM Implementation**: Instead of using off-the-shelf libraries that update Gaussian parameters sequentially, we store the parameters of all $C \times C \times K$ components as tensors in PyTorch and update them in parallel using matrix operations.
> 2. **CDF Computation**: We approximate the cumulative distribution function using the Abramowitz-Stegun formula, which allows efficient polynomial evaluation.
> 3. **Inverse CDF Computation**: We use interpolation techniques within the estimated value range, which avoids costly numerical inversion.
>
> All the above operations can be efficiently performed using simple matrix operations on tensors. Moreover, the storage of Gaussian component parameters and the additional regression head (a $1\times 1$ conv) introduced by our method are lightweight. Overall, our method demonstrates high efficiency in both training time and GPU memory, as reported in Table 13 of Appendix I. Moreover, our method introduces no additional overhead during inference.
>
> ---
>
>
> **Q2: Why use logits distributions**
>
> **A2:**  In UDA, explicit class distribution priors are often unavailable, especially in the target domain, where distribution shift is severe. Therefore, we propose leveraging the logits distributions as an online proxy to assess a model’s class-wise prediction bias.
>
> We justify this design choice from both theoretical and empirical perspectives:
>
> 1. Theoretical Justification:
>     As shown in Definition 2, the prediction bias $\mathrm{Bias}(l)$ is directly related to the probability of predicting class $l$ across all ground-truth classes, which in turn depends on the relative distributions of logits. We further show in Eq. (5) that under the assumption of independent logits distribution, this probability can be estimated by comparing the positive and negative class logits. A sufficient condition for unbiased prediction is that these distributions are aligned.
>
> 2. Empirical Evidence:
>     In Figure 1(d), we demonstrate a clear linear correlation between the prediction bias and the differences in logits distributions across classes. This supports our hypothesis that logits distributions serve as an effective proxy to capture class imbalance in the network’s behavior.
>
>
> ---
>
>
> **Q3: Alternative Distribution Estimation Methods**
>
> **A3:** Thank you for this valuable suggestion. We chose  GMMs primarily for their balance between modeling capacity and computational efficiency, which is critical for our online training setup. Specifically:
>
> - GMMs can approximate a wide range of 1D distributions with a small number of parameters.
> - They allow closed-form CDF and inverse CDF computation (using polynomial approximations), enabling efficient matrix-based implementation for large-scale training.
> - The distributions we model are scalar-valued logits for each class pair $(c, l)$, making GMMs a sufficiently expressive and computationally tractable choice.
>
> Additionally, as shown in Appendix M, GMMs empirically fit the logits distributions well, and as discussed in Appendix E, they converge quickly during training. While more complex estimators (e.g., kernel density estimation or deep density models) could be considered, they often introduce significant overhead and do not offer closed-form CDFs, which are essential for our framework.
>
>
> ---
>
> **Q4: References**
>
> **A4:** Thank you for pointing this out. These references are indeed important and representative works in UDA for semantic segmentation. We will include them in the Related Work section in the revised version and  compare with them in our experiments.
>
> ---
>
> We hope our response can resolve your concern. Please do not hesitate to let us know if you have further questions.

---

### Decision · Program_Chairs · 2025-05-01

**Decision:**

Accept (poster)

**Comment:**

All reviewers acknowledged that the paper presents solid novelty and is well-written, well-structured, and easy to follow. Initially, some concerns were raised regarding the limited scope of experiments, particularly the lack of comparisons with alternative distribution estimation methods and the fairness of existing comparisons. However, during the rebuttal phase, the authors addressed these concerns by providing additional experiments. AC has also reviewed the paper, the reviewer comments, and the rebuttal, and agrees that the motivation is clear, the writing is strong, and the experimental evaluation is thorough. Therefore, the AC recommends acceptance. It is recommended that all additional experiments and discussions from the rebuttal be incorporated into the final version of the paper.